# Quantifying and Improving Transferability in Domain Generalization

**Guojun Zhang**
School of Computer Science
University of Waterloo
Vector Institute
guojun.zhang@uwaterloo.ca

**Han Zhao**
Department of Computer Science
University of Illinois at Urbana-Champaign
hanzhao@illinois.edu

**Yaoliang Yu**
School of Computer Science
University of Waterloo
Vector Institute
yaoliang.yu@uwaterloo.ca

**Pascal Poupart**
School of Computer Science
University of Waterloo
Vector Institute
ppoupart@uwaterloo.ca

## Abstract

Out-of-distribution generalization is one of the key challenges when transferring a model from the lab to the real world. Existing efforts mostly focus on building invariant features among source and target domains. Based on invariant features, a high-performing classifier on source domains could hopefully behave equally well on a target domain. In other words, we hope the invariant features to be *transferable*. However, in practice, there are no perfectly transferable features, and some algorithms seem to learn "more transferable" features than others. How can we understand and quantify such *transferability*? In this paper, we formally define transferability that one can quantify and compute in domain generalization. We point out the difference and connection with common discrepancy measures between domains, such as total variation and Wasserstein distance. We then prove that our transferability can be estimated with enough samples and give a new upper bound for the target error based on our transferability. Empirically, we evaluate the transferability of the feature embeddings learned by existing algorithms for domain generalization. Surprisingly, we find that many algorithms are not quite learning transferable features, although few could still survive. In light of this, we propose a new algorithm for learning transferable features and test it over various benchmark datasets, including RotatedMNIST, PACS, Office-Home and WILDS-FMoW. Experimental results show that the proposed algorithm achieves consistent improvement over many state-of-the-art algorithms, corroborating our theoretical findings.[1]

## 1 Introduction

One of the cornerstone assumptions underlying the recent success of deep learning models is that the test data should share the same distribution as the training data. However, faced with ubiquitous distribution shifts in various real-world applications, such assumption hardly holds in practice. For example, a self-driving recognition system trained using data collected in the daytime may continually degrade its performance during nightfall. The system may also encounter weather or traffic conditions in a new city that never appear in the training set. In light of these potentially unseen scenarios, it is

---

[1]Code available at https://github.com/Gordon-Guojun-Zhang/Transferability-NeurIPS2021.

of paramount importance that the trained model can generalize *Out-Of-Distribution* (OOD): even if the target domain is not exactly the same as the source domain(s), the learned model should hopefully behave robustly under slight distribution shift.

To this end, one line of works focuses on learning the so-called *invariant representations* [2, 16, 57, 58]. At a colloquial level, the goal here is to learn feature embeddings that lead to indistinguishable feature distributions from different domains. In practice, both the feature embeddings and the domain discriminators are often parametrized by neural networks, leading to an adversarial game between these two. Furthermore, in order to avoid degenerate solutions, the learned features are required to be informative about the output variable as well. This is enforced by placing a predictor over the features and minimize the corresponding supervised loss simultaneously [17, 32, 48, 49].

Another line of recent works aims to learn features that can induce *invariant predictors*, first termed as the invariant risk minimization (IRM) [3, 39] paradigm. Roughly speaking, the goal of IRM is to discover a feature embedding, upon which the optimal predictors, i.e., the Bayes predictor, are invariant across the training domains. Again, at the same time, the features should be informative about the output variable as well. However, the optimization problem of IRM is rather difficult, and several follow-up works have proposed different relaxations to the original formulation [1, 26].

Despite being extensively studied, both theoretical [41, 59] and empirical [20, 23] works have shown the insufficiency of existing algorithms for domain generalization (DG). Methods based on invariant features ignore the potential shift in the marginal label distributions across domains [59] and the methods based on invariant predictors are not robust to covariate shift [26]. Perhaps surprisingly, empirical works have shown that with proper data augmentation and careful model tuning, the very basic algorithm of empirical risk minimization (ERM) demonstrates superior performance on domain generalization over existing methods on benchmark image datasets [20, 23]. This sharp gap between theory and practice calls for a fundamental understanding of the following question:

> *What kind of invariance should we look for, in order to ensure that a good model*
> *on source domains also achieves decent accuracy on a related target domain?*

In this work we attempt to answer the above question by proposing a criterion for models to look at, dubbed as *transferability*, which asks for an invariance of the *excess risks* of a predictor across domains. Different from existing proposals of invariant features and invariant predictors, which seek to find feature embeddings that respectively induce invariant marginal and conditional distributions, our notion of transferability depends on the excess risk, hence it directly takes into account the joint distribution over both the features and the labels. We show how it can be used to naturally derive a new upper bound for the target error, and then we discuss how to estimate the transferability empirically with enough samples. Our definition also inspires a method that aims to find more transferable features via representation learning using adversarial training.

Empirically, we perform experiments to measure the transferability of several existing algorithms, on both small and large scale datasets. We show that many algorithms, including ERM, are not quite transferable under the definition (Fig. 1, see more details in §5): when we go away from the optimal classifier (with distance $\delta$ in the parameter space), it could happen that the source accuracy remains high but the target accuracy drops significantly. This implies that during the training process, an existing algorithm may find a good source classifier with low target accuracy, hence violating the requirement for invariance of excess risks. In contrast, our algorithm is more transferable, and achieves consistent improvement over existing state-of-the-art algorithms, corroborating our findings.

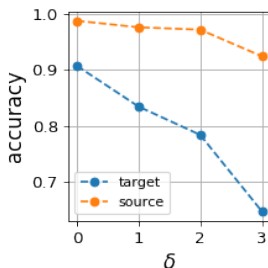

Figure 1: The target and source (test) accuracies of ERM on MNIST.

## 2  What is Transferability?

In this section we present our definition of transferability in the classification setting. The setup of domain generalization is the following:

**Settings and Notation**  Given $n$ labeled source domains $\mathcal{S}_1, \ldots, \mathcal{S}_n$, the problem of domain generalization is to learn a model from these source domains, in the hope that it performs well on an

unseen target domain $\mathcal{T}$ that is "similar" to the source domains. Throughout the paper, we assume that both the source domains and the unseen target domain share the same input and output spaces, denoted as $\mathcal{X}$ and $\mathcal{Y}$, respectively. For multi-class classification, the output space $\mathcal{Y} = [K]$ is a set of labels for multi-class classification. For binary classification, we consider $\mathcal{Y} = \{-1, +1\}$. Denote $\mathcal{H}$ as the hypothesis class. We define the classification error of a classifier $h \in \mathcal{H}$ on a domain $\mathcal{D}$ (or $\mathcal{S}$ for source domains, or $\mathcal{T}$ for target domains) as:[2]

$$\epsilon_{\mathcal{D}}(h) = \mathbb{E}_{(x,y)\sim\mathcal{D}}[\ell(h(x), y)]. \tag{1}$$

For $\ell(h(x), y) = \mathbb{1}(h(x) \neq y)$, where $\mathbb{1}(\cdot)$ is the usual indicator function, we use $\epsilon_{\mathcal{D}}^{0-1}(h)$ to denote it is the 0-1 loss.

In domain generalization, we often have several source domains. For the ease of presentation, we only consider a single source domain in this section, and later extend to the general case in Section 5. Given two domains, the source domain $\mathcal{S}$ and the target domain $\mathcal{T}$, the task of domain generalization is to transfer a classifier $h$ that performs well on $\mathcal{S}$ to $\mathcal{T}$. We ask: how much of the success of $h$ on $\mathcal{S}$ can be transferred to $\mathcal{T}$?

Note that in order to evaluate the transferability from $\mathcal{S}$ to $\mathcal{T}$, we need information from the target domain, similar to the test phase in traditional supervised learning. We believe a good criterion of transferability should satisfy the following properties:

1. *Quantifiable:* the notion should be quantifiable and can be computed in practice;
2. Any near-optimal source classifier should be near-optimal on the target domain.
3. If the two domains are similar, as measured by e.g., total variation, then they are transferable to each other, but the converse may not be true.

At first glance the second criterion above might seem too strong and restrictive. However, we argue that in the task of domain generalization, we only have labeled source data and there is no clue to distinguish a classifier from another if both of them perform equally well on the source domain. Based on the second property, we first propose the following definition of transferability:

**Definition 1 (transferability).** *$\mathcal{S}$ is $(\delta_{\mathcal{S}}, \delta_{\mathcal{T}})_{\mathcal{H}}$-transferable to $\mathcal{T}$ if for $\delta_{\mathcal{S}} > 0$, there exists $\delta_{\mathcal{T}} > 0$ such that* $\mathrm{argmin}(\epsilon_{\mathcal{S}}, \delta_{\mathcal{S}})_{\mathcal{H}} \subseteq \mathrm{argmin}(\epsilon_{\mathcal{T}}, \delta_{\mathcal{T}})_{\mathcal{H}}$, *where:*

$$\mathrm{argmin}(\epsilon_{\mathcal{D}}, \delta_{\mathcal{D}})_{\mathcal{H}} := \{h \in \mathcal{H} : \epsilon_{\mathcal{D}}(h) \leq \inf_{h \in \mathcal{H}} \epsilon_{\mathcal{D}}(h) + \delta_{\mathcal{D}}\}.$$

In the literature the set $\mathrm{argmin}(\epsilon_{\mathcal{D}}, \delta_{\mathcal{D}})_{\mathcal{H}}$ is also known as a $\delta_{\mathcal{D}}$-*minimal set* [24] of $\epsilon_{\mathcal{D}}$, which represents the near-optimal set of classifiers. Note that the $\delta$-minimal set depends on the hypothesis class $\mathcal{H}$. Throughout the paper, we omit the subscript $\mathcal{H}$ in the definition when there is no confusion. Def. 1 says that near-optimal source classifiers are also near-optimal target classifiers. Furthermore, it is easy to verify that our transferability is transitive: if $\mathcal{S}$ is $(\delta_{\mathcal{S}}, \delta_{\mathcal{P}})$-transferable to $\mathcal{P}$, and $\mathcal{P}$ is $(\delta_{\mathcal{P}}, \delta_{\mathcal{T}})$-transferable to $\mathcal{T}$, then $\mathcal{S}$ is $(\delta_{\mathcal{S}}, \delta_{\mathcal{T}})$-transferable to $\mathcal{T}$.

Next we define transfer measures, which we will show to be equivalent with Def. 1 in Prop. 5.

**Definition 2 (quantifiable transfer measures).** *Given some $\Gamma \subseteq \mathcal{H}$, $\epsilon_{\mathcal{S}}^* := \inf_{h\in\Gamma} \epsilon_{\mathcal{S}}(h)$ and $\epsilon_{\mathcal{T}}^* := \inf_{h\in\Gamma} \epsilon_{\mathcal{T}}(h)$ we define the one-sided transfer measure, symmetric transfer measure and the realizable transfer measure respectively as:*

$$\mathrm{T}_{\Gamma}(\mathcal{S}\|\mathcal{T}) := \sup_{h\in\Gamma} \epsilon_{\mathcal{T}}(h) - \epsilon_{\mathcal{T}}^* - (\epsilon_{\mathcal{S}}(h) - \epsilon_{\mathcal{S}}^*), \tag{2}$$

$$\mathrm{T}_{\Gamma}(\mathcal{S}, \mathcal{T}) := \max\{\mathrm{T}_{\Gamma}(\mathcal{S}\|\mathcal{T}), \mathrm{T}_{\Gamma}(\mathcal{T}\|\mathcal{S})\} = \sup_{h\in\Gamma} |\epsilon_{\mathcal{S}}(h) - \epsilon_{\mathcal{S}}^* - (\epsilon_{\mathcal{T}}(h) - \epsilon_{\mathcal{T}}^*)|, \tag{3}$$

$$\mathrm{T}_{\Gamma}^{\mathrm{r}}(\mathcal{S}, \mathcal{T}) := \sup_{h\in\Gamma} |\epsilon_{\mathcal{S}}(h) - \epsilon_{\mathcal{T}}(h)|. \tag{4}$$

The distinction between $\Gamma$ and $\mathcal{H}$ will become apparent in Prop. 5. Note that the one-sided transfer measure is not symmetric. If we want the two domains $\mathcal{S}$ and $\mathcal{T}$ to be mutually transferable to each other, we can use the symmetric transfer measure. We call both quantities as *transfer measures*. Furthermore, the symmetric transfer measure reduces to (4) in the realizable case when $\epsilon_{\mathcal{S}}^* = \epsilon_{\mathcal{T}}^* = 0$. In statistical learning theory, $\epsilon_{\mathcal{D}}(h) - \epsilon_{\mathcal{D}}^*$ is often known as an *excess risk* [24], which is the relative error compared to the optimal classifier. The transfer measures can thus be represented with the difference of excess risks. With Def. 2, we can immediately obtain the following result that upper bounds the target error:

---

[2]Throughout the paper, we will use the terms domain and distribution interchangeably.

**Proposition 3** (**target error bound**). *Given* $\Gamma \subseteq \mathcal{H}$, *for any* $h \in \Gamma$, *the target error is bounded by:*

$$\epsilon_{\mathcal{T}}(h) \leq \epsilon_{\mathcal{S}}(h) + \epsilon_{\mathcal{T}}^* - \epsilon_{\mathcal{S}}^* + \mathrm{T}_\Gamma(\mathcal{S}\|\mathcal{T}) \leq \epsilon_{\mathcal{S}}(h) + \epsilon_{\mathcal{T}}^* - \epsilon_{\mathcal{S}}^* + \mathrm{T}_\Gamma(\mathcal{S}, \mathcal{T}). \tag{5}$$

The first error bound of such type for a target domain uses $\mathcal{H}$-divergence [8, 9, 12] for binary classification (or more rigorously, the $\mathcal{H}\Delta\mathcal{H}$-divergence). The main difference between ours and $\mathcal{H}$-divergence is that $\mathcal{H}$-divergence only concerns about the marginal input distributions, whereas the transfer measures depend on the joint distributions over both the inputs and the labels. We note that Proposition 3 is general and works in the multi-class case as well. Moreover, even in the binary classification case we can prove that our Proposition 3 is tighter than $\mathcal{H}$-divergence (see Proposition 27 in the appendix).

In practice we may not know the optimal errors. In this case, we can use the realizable transfer measure to upper bound the symmetric transfer measure (note that $\epsilon_{\mathcal{S}}^*$ or $\epsilon_{\mathcal{T}}^*$ may not be zero):

**Proposition 4.** *For* $\Gamma \subseteq \mathcal{H}$ *and domains* $\mathcal{S}, \mathcal{T}$ *we have:* $\mathrm{T}_\Gamma(\mathcal{S}, \mathcal{T}) \leq 2\mathrm{T}_\Gamma^{\mathrm{r}}(\mathcal{S}, \mathcal{T})$.

Since Def. 1 essentially asks that the excess risks of approximately optimal classifiers on the source domain are comparable between the source and target domains, we can show that Def. 1 and Def. 2 are equivalent if $\Gamma$ is a $\delta$-minimal set:

**Proposition 5** (**equivalence between transferability and transfer measures**). *Let* $\delta_{\mathcal{S}} > 0$ *and* $\Gamma = \mathrm{argmin}(\epsilon_{\mathcal{S}}, \delta_{\mathcal{S}})$ *and suppose* $\inf_{h \in \Gamma} \epsilon_{\mathcal{T}}(h) = \inf_{h \in \mathcal{H}} \epsilon_{\mathcal{T}}(h)$. *If* $\mathrm{T}_\Gamma(\mathcal{S}\|\mathcal{T}) \leq \delta$ *or* $\mathrm{T}_\Gamma(\mathcal{S}, \mathcal{T}) \leq \delta$, *then* $\mathcal{S}$ *is* $(\delta_{\mathcal{S}}, \delta + \delta_{\mathcal{S}})$-*transferable to* $\mathcal{T}$. *Furthermore, if* $\mathcal{S}$ *is* $(\delta_{\mathcal{S}}, \delta_{\mathcal{T}})$-*transferable to* $\mathcal{T}$, *then* $\mathrm{T}_\Gamma(\mathcal{S}\|\mathcal{T}) \leq \delta_{\mathcal{T}}$ *and* $\mathrm{T}_\Gamma(\mathcal{S}, \mathcal{T}) \leq \max\{\delta_{\mathcal{S}}, \delta_{\mathcal{T}}\}$.

In Prop. 5, we do not require $\Gamma = \mathcal{H}$ since it is unnecessary to impose that all classifiers in $\mathcal{H}$ have similar excess risks on source and target domains. Instead, we only constrain $\Gamma$ to be a $\delta$-minimal set, i.e., $\Gamma$ includes approximately optimal classifiers of $\mathcal{S}$. See also Example 8. An additional assumption is that $\Gamma$ also includes the optimal classifier of $\mathcal{T}$ which can be ensured by controlling $\delta_{\mathcal{S}}$.

## 2.1 Comparison with other discrepancy measures between domains

In this subsection, we compare the realizable transfer measure (4) with other discrepancy measures between domains and focus on the 0-1 loss $\epsilon_{\mathcal{D}}^{0-1}$. We first note that $\mathrm{T}_\Gamma^{\mathrm{r}}(\mathcal{S}, \mathcal{T})$ can be written as an integral probability metric (IPM) [36, 46]. The l.h.s. of (4) can be written as:

$$\mathrm{T}_\Gamma^{\mathrm{r}}(\mathcal{S}, \mathcal{T}) := d_{\mathcal{F}_\Gamma}(\mathcal{S}, \mathcal{T}), \text{ where } d_{\mathcal{F}}(\mathcal{S}, \mathcal{T}) = \sup_{f \in \mathcal{F}} \left| \sum_y \int f(x, y)(p_{\mathcal{S}}(x, y) - p_{\mathcal{T}}(x, y)) dx \right|, \tag{6}$$

and $\mathcal{F}_\Gamma := \{(x, y) \mapsto \mathbb{1}(h(x) \neq y), h \in \Gamma\}$. Typical IPMs [46] include MMD, Wasserstein distance, Dudley metric and the Kolmogorov–Smirnov distance (see Appendix B.2 for more details). However, $\mathcal{F}_\Gamma$ is fundamentally different from these IPMs since it relies on an underlying function class $\Gamma$. Our realizable transfer measure shares some similarity with Arora et al. [4], where a changeable function class is used, but the exact choices of the function class are different.

Even though the transferability can be written in terms of IPM, it is in fact a pseudo-metric:

**Proposition 6** (**pseudo-metric**). *For a general loss* $\epsilon_{\mathcal{D}}$ *as in* (1), $\mathrm{T}_\Gamma^{\mathrm{r}}(\mathcal{S}, \mathcal{T})$ *is a pseudo-metric, i.e., for any distributions* $\mathcal{S}, \mathcal{T}, \mathcal{P}$ *on the same underlying space, we have* $\mathrm{T}_\Gamma^{\mathrm{r}}(\mathcal{S}, \mathcal{S}) = 0$, $\mathrm{T}_\Gamma^{\mathrm{r}}(\mathcal{S}, \mathcal{T}) = \mathrm{T}_\Gamma^{\mathrm{r}}(\mathcal{T}, \mathcal{S})$ *(symmetry), and* $\mathrm{T}_\Gamma^{\mathrm{r}}(\mathcal{S}, \mathcal{T}) \leq \mathrm{T}_\Gamma^{\mathrm{r}}(\mathcal{S}, \mathcal{P}) + \mathrm{T}_\Gamma^{\mathrm{r}}(\mathcal{P}, \mathcal{T})$ *(triangle inequality).*

However in general $\mathrm{T}_\Gamma^{\mathrm{r}}(\mathcal{S}, \mathcal{T})$ is not a metric since $\mathrm{T}_\Gamma^{\mathrm{r}}(\mathcal{S}, \mathcal{T}) = 0$ even if $\mathcal{S} \neq \mathcal{T}$. For instance, taking $\Gamma = \{h^*\}$ to be the optimal classifier on both $\mathcal{S}$ and $\mathcal{T}$. we have $\mathrm{T}_\Gamma^{\mathrm{r}}(\mathcal{S}, \mathcal{T}) = 0$, but $\mathcal{S}$ and $\mathcal{T}$ could differ a lot (see Figure 2). In the next result we discuss the connection between realizable transfer measures and total variation (c.f. Appendix B.2).

**Proposition 7** (**equivalence with total variation**). *For binary classification with labels* $\{-1, 1\}$, *given the 0-1 loss* $\epsilon_{\mathcal{D}} = \epsilon_{\mathcal{D}}^{0-1}$, *we have* $\mathrm{T}_\Gamma^{\mathrm{r}}(\mathcal{S}, \mathcal{T}) \leq d_{\mathrm{TV}}(\mathcal{S}, \mathcal{T})$ *for domains* $\mathcal{S}, \mathcal{T}$ *and any* $\Gamma \subseteq \mathcal{H}$. *Denote* $\mathcal{H}_t$ *to be the set of all binary classifiers. Then we have* $d_{\mathrm{TV}}(\mathcal{S}, \mathcal{T}) \leq 4\mathrm{T}_{\mathcal{H}_t}^{\mathrm{r}}(\mathcal{S}, \mathcal{T})$.

Prop. 7 tells us that transfer measures (see also Prop. 4) are no stronger than total variation, and in the realizable case, (3) is equivalent to the similarity of domains (as measured by total variation) if $\Gamma$ is unconstrained. We can moreover show that transfer measures are strictly weaker, if we choose $\Gamma$ to be some $\delta$-minimal set:

**Example 8** (**very dissimilar joint distributions but transferable**). *We study the distributions described in Figure 2. The joint distributions are very dissimilar, i.e., for any $X, Y$ in the domain, $|p_{\mathcal{S}}(X, Y) - p_{\mathcal{T}}(X, Y)| = 0.8$. Define*

$$h_{\rho}(X) = \begin{cases} 1 & if -1 \leq X < \rho \\ -1 & if \rho \leq X < 1 \end{cases}. \tag{7}$$

*We choose the hypothesis class $\mathcal{H} = \{h_{\rho}, \rho \in [-1, 1]\}$ and $\Gamma = \{h_{\rho}, |\rho| \leq \delta/0.8\}$ (for small $\delta$, say $\delta < 0.01$) to be some neighborhood of the optimal source classifier $h^* = h_0$. Then $\mathrm{T}_{\Gamma}(\mathcal{S}, \mathcal{T}) = \sup_{h \in \Gamma} |\epsilon_{\mathcal{S}}(h) - \epsilon_{\mathcal{T}}(h)| = \delta$, and $\mathcal{S}$ is $(\delta_{\mathcal{S}}, \delta + \delta_{\mathcal{S}})$-transferable to $\mathcal{T}$ on $\Gamma$ for any $\delta_{\mathcal{S}} > 0$ according to Prop. 5. Note that $\epsilon_{\mathcal{S}}^* = \epsilon_{\mathcal{T}}^* = 0$.*

## 3 Computing Transferability

In the last section we proposed a new concept called transferability. However, although Def. 1 provides a theoretically sound result for transferability, it is hard to verify it in practice, since we cannot exhaust all approximately good classifiers, especially for rich models such as deep neural networks. Nevertheless, Prop. 3 and Prop. 5 provide a framework to compute transferability through transfer measures, despite their simplicity. In this section we discuss how to compute these quantities by making necessary approximations based on transfer measures. There are two difficulties we need to overcome: **(1)** In practice we only have finite samples drawn from true distributions; **(2)** We need a surrogate loss such as cross entropy for training and the 0-1 loss for evaluation. In §3.1 we show that our transfer measures can be estimated with enough samples, and in §3.2 we discuss transferability with a surrogate loss. These results will be used in our algorithms in the next section.

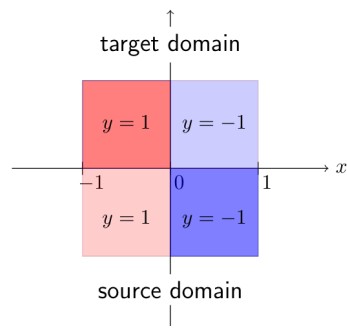

Figure 2: Visualization of Example 8. **Source domain**: $P_{\mathcal{S}}(Y = 1, -1 \leq X < 0) = 0.1$, $P_{\mathcal{S}}(Y = -1, 0 \leq X < 1) = 0.9$. **Target domain**: $P_{\mathcal{T}}(Y = 1, -1 \leq X < 0) = 0.9$, $P_{\mathcal{T}}(Y = -1, 0 \leq X < 1) = 0.1$. The dark and light colors show the intensity of the probability mass. The vertical axis denotes whether it is the target or source domain (above or below $x$-axis).

### 3.1 Estimation of transferability

We show how to estimate the transfer measure $\mathrm{T}_{\Gamma}(\mathcal{S} \| \mathcal{T})$ from finite samples. Other versions of transfer measures in Def. 1 follow analogously (see Appendix A for more details).

**Lemma 9** (**reduction of estimation error**). *Given general loss $\epsilon_{\mathcal{D}}$ as in (1), suppose $\widehat{\mathcal{S}}$ and $\widehat{\mathcal{T}}$ are i.i.d. sample distributions drawn from distributions of $\mathcal{S}$ and $\mathcal{T}$, then for any $\Gamma \subseteq \mathcal{H}$ we have:*

$$\mathrm{T}_{\Gamma}(\mathcal{S} \| \mathcal{T}) \leq \mathrm{T}_{\Gamma}(\widehat{\mathcal{S}} \| \widehat{\mathcal{T}}) + 2\mathrm{est}_{\Gamma}(\mathcal{S}) + 2\mathrm{est}_{\Gamma}(\mathcal{T}),$$

*with the estimation errors $\mathrm{est}_{\Gamma}(\mathcal{S}) = \sup_{h \in \Gamma} |\epsilon_{\mathcal{S}}(h) - \epsilon_{\widehat{\mathcal{S}}}(h)|$, $\mathrm{est}_{\Gamma}(\mathcal{T}) = \sup_{h \in \Gamma} |\epsilon_{\mathcal{T}}(h) - \epsilon_{\widehat{\mathcal{T}}}(h)|$.*

This lemma tells us that estimating transferability is no harder than computing the estimation errors of both domains. If the function class $\Gamma$ has uniform convergence property [44], then we can guarantee efficient estimation of transferability. We first bound the sample complexity through Rademacher complexity, which is a standard tool in bounding estimation errors [5]:

**Theorem 10** (**estimation error with Rademacher complexity**). *Given the 0-1 loss $\epsilon_{\mathcal{D}} = \epsilon_{\mathcal{D}}^{0-1}$, suppose $\widehat{\mathcal{S}}$ and $\widehat{\mathcal{T}}$ are sample sets with $m$ and $k$ samples drawn i.i.d. from distributions $\mathcal{S}$ and $\mathcal{T}$, respectively. For any $\Gamma \subseteq \mathcal{H}$ the following holds with probability $1 - \delta$:*

$$\mathrm{T}_{\Gamma}(\mathcal{S} \| \mathcal{T}) \leq \mathrm{T}_{\Gamma}(\widehat{\mathcal{S}} \| \widehat{\mathcal{T}}) + 4\mathfrak{R}_m(\mathcal{F}_{\Gamma}) + 4\mathfrak{R}_k(\mathcal{F}_{\Gamma}) + 2\sqrt{\frac{\log(4/\delta)}{2m}} + 2\sqrt{\frac{\log(4/\delta)}{2k}},$$

*where $\mathcal{F}_{\Gamma} := \{(x, y) \mapsto \mathbb{1}(h(x) \neq y), h \in \Gamma\}$. If furthermore, $\Gamma$ is a set of binary classifiers with labels $\{-1, 1\}$, then $2\mathfrak{R}_m(\mathcal{F}_{\Gamma}) = \mathfrak{R}_m(\Gamma)$, $2\mathfrak{R}_k(\mathcal{F}_{\Gamma}) = \mathfrak{R}_k(\Gamma)$.*

We also provide estimation error results using Vapnik–Chervonenkis (VC) dimension and Natarajan dimension in Appendix B.3. It is worth mentioning that the VC dimension of piecewise-polynomial neural networks has been upper bounded in Bartlett et al. [7]. Since transfer measures can be estimated, in later sections we do not distinguish the sample sets $\widehat{\mathcal{S}}, \widehat{\mathcal{T}}$ and the underlying distributions $\mathcal{S}, \mathcal{T}$.

## 3.2 Transferability with a surrogate loss

Due to the intractability of minimizing the 0-1 loss, we need to use a surrogate loss [6] for training in practice. In this section, we discuss this nuance w.r.t. transferability. We will focus on the most commonly used surrogate loss, cross entropy (CE), although some of the results can be easily adapted to other loss functions. To distinguish a surrogate loss from the 0-1 loss, we use $\epsilon_{\mathcal{D}}$ from now on for a surrogate loss and $\epsilon_{\mathcal{D}}^{0-1}$ for the 0-1 loss. One of the difficulties is the non-equivalence between $\delta$-minimal sets w.r.t. the 0-1 loss and a surrogate loss, i.e. $\mathrm{argmin}(\epsilon_{\mathcal{D}}, \delta_{\mathcal{D}})$ might be quite different from $\mathrm{argmin}(\epsilon_{\mathcal{D}}^{0-1}, \delta_{\mathcal{D}})$. Moreover, it is not practical to find all elements in $\mathrm{argmin}(\epsilon_{\mathcal{D}}^{0-1}, \delta_{\mathcal{D}})$ since the loss is nonconvex and nonsmooth. In light of these difficulties, we propose a more practical notion of transferability based on surrogate loss $\epsilon_{\mathcal{D}}$:

**Proposition 11** (**transfer measure with a surrogate loss**). *Given surrogate loss $\epsilon_{\mathcal{D}} \geq \epsilon_{\mathcal{D}}^{0-1}$ on a general domain $\mathcal{D}$. Suppose $\Gamma = \mathrm{argmin}(\epsilon_{\mathcal{S}}, \delta_{\mathcal{S}})$ and denote $\epsilon_{\mathcal{T}}^* = \inf_{h \in \Gamma} \epsilon_{\mathcal{T}}(h)$, $\epsilon_{\mathcal{S}}^* = \inf_{h \in \Gamma} \epsilon_{\mathcal{S}}(h)$, $(\epsilon_{\mathcal{T}}^{0-1})^* = \inf_{h \in \mathcal{H}} \epsilon_{\mathcal{T}}^{0-1}(h)$. If the following holds:*

$$\mathrm{T}_{\Gamma}(\mathcal{S}\|\mathcal{T}) = \sup_{h \in \Gamma} \epsilon_{\mathcal{T}}(h) - \epsilon_{\mathcal{T}}^* - (\epsilon_{\mathcal{S}}(h) - \epsilon_{\mathcal{S}}^*) \leq \delta, \tag{8}$$

*then we have $\mathrm{argmin}(\epsilon_{\mathcal{S}}, \delta_{\mathcal{S}}) \subseteq \mathrm{argmin}(\epsilon_{\mathcal{T}}^{0-1}, \delta + \delta_{\mathcal{S}} + \epsilon_{\mathcal{T}}^* - (\epsilon_{\mathcal{T}}^{0-1})^*)$.*

This proposition implies that if the transfer measure is small, then a near-optimal classifier of the surrogate loss in the source domain would be near-optimal in the target domain for the 0-1 loss. It also gives us a practical framework to guarantee transferability, which we will discuss in more depth in Section 4. Assume $\epsilon_{\mathcal{D}} : \mathcal{H} \to \mathbb{R}$ to be Lipschitz continuous and strongly convex, which is satisfied for the cross entropy loss (see Appendix B.4). We are able to translate the $\delta$-minimal set to $L_p$ balls in the function space:

$$C_1 \|h - h^*\|_{2,\mathcal{D}} \leq \epsilon_{\mathcal{D}}(h) - \epsilon_{\mathcal{D}}(h^*) \leq C_2 \|h - h^*\|_{1,\mathcal{D}}, \tag{9}$$

where $C_1$ and $C_2$ are absolute constants and $h^*$ is an optimal classifier. The function norms $\|\cdot\|_{1,\mathcal{D}}$ and $\|\cdot\|_{2,\mathcal{D}}$ are the usual $L_p$ norms over distribution $\mathcal{D}$. Since the classifier $h = q(\theta, \cdot)$ is usually parameterized with, say a neural network, we further upper bound the function norms by the distance of parameters, that is, for $1 \leq p < \infty$, $h = q(\theta, \cdot)$ and $h' = q(\theta', \cdot)$, we have $\|h - h'\|_{p,\mathcal{D}} \leq L\|\theta - \theta'\|_2$, with $L$ some Lipschitz constant of $q$ (Appendix B.4). Combined with (9), we obtain:

$$\epsilon_{\mathcal{D}}(h) - \epsilon_{\mathcal{D}}(h') \leq LC_2 \|\theta - \theta'\|_2. \tag{10}$$

In other words, if the parameters are close enough, then the losses should not differ too much. We denote $\|\cdot\|_2$ as the Euclidean norm, and for later convenience we will omit the subscript in $\|\cdot\|_2$.

# 4 Algorithms for Evaluating and Improving Transferability

The notion of transferability is defined w.r.t. domains, hence by learning feature embeddings that induce certain feature distributions, one can aim to improve transferability of two given domains. In this section we design algorithms to evaluate and improve transferability by learning such transformations. To start with, let $g : \mathcal{X} \to \mathcal{Z}$ be a feature embedding (a.k.a. featurizer), where $\mathcal{Z}$ is understood to be a feature space. By a joint distribution $\mathcal{D}^g$ (or $\mathcal{S}^g$, $\mathcal{T}^g$) we mean a distribution on $g(\mathcal{X}) \times \mathcal{Y}$. Formally, we are dealing with push-forwards of distributions:

$$\mathcal{S}^g := (g, \mathrm{id})\#\mathcal{S}, \ \mathcal{T}^g := (g, \mathrm{id})\#\mathcal{T}, \tag{11}$$

where $(g, \mathrm{id}) : (x, y) \mapsto (g(x), y)$ is a function on $\mathcal{X} \times \mathcal{Y}$. $\mathcal{S}$ and $\mathcal{T}$ here are joint distributions on $\mathcal{X} \times \mathcal{Y}$, and here we specify $\mathcal{X}$ to be the space of the original signal such as an image. Since $\mathcal{S}$ and $\mathcal{T}$ cannot be changed, what we are evaluating here is the feature embedding $g$. The key quantity is transfer measures as in (8):

$$\mathrm{T}_{\Gamma}(\mathcal{S}^g\|\mathcal{T}^g) = \sup_{h \in \Gamma} \epsilon_{\mathcal{T}^g}(h) - \epsilon_{\mathcal{T}^g}^* - (\epsilon_{\mathcal{S}^g}(h) - \epsilon_{\mathcal{S}^g}^*), \quad \Gamma = \mathrm{argmin}(\epsilon_{\mathcal{S}^g}, \delta_{\mathcal{S}^g}). \tag{12}$$

Although $\Gamma$ is hard to compute, we can use (10) to obtain a lower bound of (12). That is, given a parametrization of the classifier $h = q(\theta, \cdot)$ and the optimal classifier $h^* = q(\theta^*, \cdot)$, we have:

$$
\begin{aligned}
\mathrm{T}_\Gamma(\mathcal{S}^g \| \mathcal{T}^g) &\geq \sup_{\|\theta - \theta^*\| \leq \delta} \epsilon_{\mathcal{T}^g}(h) - \epsilon_{\mathcal{S}^g}(h) - \epsilon^*_{\mathcal{T}^g} + \epsilon^*_{\mathcal{S}^g} \\
&\geq \sup_{\|\theta - \theta^*\| \leq \delta} \epsilon_{\mathcal{T}^g}(h) - \epsilon_{\mathcal{S}^g}(h) - \epsilon_{\mathcal{T}^g}(\widehat{h^*}) \\
&\approx \sup_{\|\theta - \widehat{\theta^*}\| \leq \delta} \epsilon_{\mathcal{T}^g}(h) - \epsilon_{\mathcal{S}^g}(h) - \epsilon_{\mathcal{T}^g}(\widehat{h^*}) \quad (13)
\end{aligned}
$$

where $\delta > 0$ depends on $\Gamma$ and the constant in (10). In the second and the third lines, we approximated the optimal errors $\epsilon^*_{\mathcal{T}^g}$ and $\epsilon^*_{\mathcal{S}^g}$ with $0 \leq \epsilon^*_{\mathcal{S}^g} \leq \epsilon_{\mathcal{S}^g}(\widehat{h^*})$, $0 \leq \epsilon^*_{\mathcal{T}^g} \leq \epsilon_{\mathcal{T}^g}(\widehat{h^*})$, and we use the learned classifier $\widehat{h^*} = q(\widehat{\theta^*}, \cdot)$ as a surrogate for the optimal classifier. As a result, if the r.h.s. of (13) is large, then $\mathcal{S}^g$ is not quite transferable to $\mathcal{T}^g$.

---

**Algorithm 1:** Algorithm for evaluating transferability among multiple domains

---

**Input:** learned feature embedding $g$, learned classifier $\widehat{h^*} = q(\widehat{\theta^*}, \cdot)$, target sample training set
  $\mathcal{T} = \mathcal{S}_0$, sample training sets $\mathcal{S}_1, \ldots, \mathcal{S}_n$, ascent optimizer, minimal errors $\epsilon^*_{\mathcal{S}_i} \approx \epsilon_{\mathcal{S}_i}(\widehat{h^*})$,
  adversarial radius $\delta$
**Initialize:** a classifier $h = q(\theta, \cdot)$ and $\theta = \widehat{\theta^*}$, gap $= -\infty$
**for** $t$ *in* $1 \ldots T$ **do**
  Find $\max_i \epsilon_{\mathcal{S}_i}(h \circ g)$ and $\min_i \epsilon_{\mathcal{S}_i}(h \circ g)$ and corresponding indices $j$ and $k$
  Run an ascent optimizer on $h$ to maximize $\mathrm{gap}_0 = \epsilon_{\mathcal{S}_j}(h \circ g) - \epsilon_{\mathcal{S}_k}(h \circ g)$
  Project $\theta$ onto the Euclidean ball $\|\theta - \widehat{\theta^*}\| \leq \delta$
  **if** $\mathrm{gap}_0 > \mathrm{gap}$ **then**
    $\lfloor$ gap $= \mathrm{gap}_0$, save accuracies and losses of each domain

**Output:** $j, k, h, \epsilon_{\mathcal{S}_j}(h \circ g) - \epsilon_{\mathcal{S}_k}(h \circ g), \epsilon_{\mathcal{S}_j}(\widehat{h^*}), \epsilon_{\mathcal{S}_k}(\widehat{h^*})$

---

We can thus design an algorithm to evaluate the transferability in Section 4.1. By computing the lower bound in (13), we can disprove the transferability as in Prop. 5 and Prop. 11. Computing the lower bound in (13) can be regarded as an attack method: there is an adversary trying to show that $\mathcal{S}^g$ is not transferable to $\mathcal{T}^g$. For this attack, we could also design a defence method aiming to minimize the lower bound and learn more transferable features.

### 4.1 Algorithm for evaluating transferability

In domain generalization we have one target domain and more than one source domains. To ease the presentation, we denote $\mathcal{S}_0 = \mathcal{T}$ (and thus $\mathcal{S}_0^g = \mathcal{T}^g$) and extend the index set to be $\{0, 1, \cdots, n\}$. We need to evaluate the transferability (13) between all pairs of $\mathcal{S}_i^g$ and $\mathcal{S}_j^g$. Algorithm 1 gives an efficient method to compute the worst-case gap $\sup_{\|\theta - \widehat{\theta^*}\| \leq \delta} \epsilon_{\mathcal{S}_i^g}(h) - \epsilon_{\mathcal{S}_j^g}(h)$ among all pairs of $(i, j)$. Essentially, it finds the worst pair of $(i, j)$ at each step such that the gap $\epsilon_{\mathcal{S}_i^g}(h) - \epsilon_{\mathcal{S}_j^g}(h)$ takes the largest value, and then maximize this gap over parameter $\theta$ through gradient ascent.

Note that the computation of (13) also depends on the information from the target domain. This is valid since we are only *evaluating* but not *training* over these domains.

### 4.2 Algorithm for improving transferability

The evaluation sub-procedure provides us a way to pick a pair of non-transferable domains $(\mathcal{S}_i^g, \mathcal{S}_j^g)$, which in turn could be used to improve the transferability among all source domains by updating the feature embedding $g$ such that the gap $\sup_{\|\theta - \theta^*\| \leq \delta} \epsilon_{\mathcal{S}_i^g}(h) - \epsilon_{\mathcal{S}_j^g}(h)$ for $(i, j) \in [n] \times [n]$. Simultaneously, we also require that the feature embedding $g$ preserves information for the target task of interest. With the parametrization $h = q(\theta, \cdot)$, $h' = q(\theta', \cdot)$, the overall optimization problem can be formulated as:

$$
\min_{g, h} \max_{\|\theta' - \theta\| \leq \delta} \frac{1}{n} \sum_{i=1}^{n} \epsilon_{\mathcal{S}_i}(h \circ g) + \left( \max_i \epsilon_{\mathcal{S}_i}(h' \circ g) - \min_i \epsilon_{\mathcal{S}_i}(h' \circ g) \right). \quad (14)
$$

Intuitively, we want to learn a common feature embedding and a classifier such that all source errors are small and the pairwise transferability between source domains is also small. If the optimization problem is properly solved, then we have the following guarantee:

**Theorem 12** (**optimization guarantee**). *Assume that the function $q(\cdot, x)$ is $L_\theta$ Lipschitz continuous for any $x$. Suppose we have learned a feature embedding $g$ and a classifier $h$ such that the loss functional $\epsilon_{\mathcal{S}_i^g} : \mathcal{H} \to \mathbb{R}$ is $L_\ell$ Lipschitz continuous w.r.t. distribution $\mathcal{S}_i^g$ for $i \in [n]$ and*

$$\max_{\|\theta' - \theta\| \leq \delta} \frac{1}{n} \sum_{i=1}^{n} \epsilon_{\mathcal{S}_i}(h \circ g) + \left(\max_i \epsilon_{\mathcal{S}_i}(h' \circ g) - \min_i \epsilon_{\mathcal{S}_i}(h' \circ g)\right) \leq \eta, \tag{15}$$

*where $\theta, \theta'$ are parameters of $h$ and $h'$. Then for any $h' \in \Gamma = \{q(\theta', \cdot) : \|\theta - \theta'\| \leq \delta\}$, we have:*

$$\mathtt{T}_\Gamma^\mathtt{r}(\mathcal{T}_1^g, \mathcal{T}_2^g) \leq \eta, \quad \epsilon_{\mathcal{S}_i}(h' \circ g) \leq \eta + L_\ell L_\theta \delta, \quad \epsilon_{\mathcal{T}}(h' \circ g) \leq 2\eta + L_\ell L_\theta \delta, \tag{16}$$

*for any $\mathcal{T}_1^g, \mathcal{T}_2^g, \mathcal{T}^g \in \mathrm{conv}(\mathcal{S}_1^g, \dots, \mathcal{S}_n^g)$ and any $i \in [n]$.*

The Lipschitzness assumption for $\epsilon_{\mathcal{S}_i^g}$ is mild and can be satisfied for cross entropy loss (c.f. Appendix B.4.1). Here $\mathrm{conv}(\cdot)$ denotes the convex hull in the same sense as Albuquerque et al. [2], i.e., each element is a mixture of source distributions. Thm 12 tells us that if we can solve the optimization problem (14) properly, we can guarantee transferability on a neighborhood of the classifier, as an approximation of the $\delta$-minimal set. We thus propose Algorithm 2, which shares similarity with existing frameworks, such as DANN [16] and Distributional Robust Optimization [43, 45], in the sense that they all involve adversarial training and minimax optimization. However, the objective in our case is different and we provide a more detailed comparison with existing methods in Appendix B.5.

---

**Algorithm 2:** Transfer algorithm for domain generalization

---

**Input:** samples sets of source domains $\mathcal{S}_1, \dots, \mathcal{S}_n$, feature embedding $g$, classifier $h = q(\theta, \cdot)$, adversarial classifier $h' = q(\theta', \cdot)$, surrogate loss $\epsilon_\mathcal{D}$, adversarial radius $\delta$, ascent optimizer, descent optimizer, weight parameter $\lambda$, number of epochs $T$

**for** $t$ *in* $1 \dots T$ **do**

    Compute $\max_i \epsilon_{\mathcal{S}_i}(h \circ g)$ and $\min_i \epsilon_{\mathcal{S}_i}(h \circ g)$

    Initialization $h' = h$ (or $\theta' = \theta$)

    **for** $k$ *in* $1 \dots N$ **do**

        Run the ascent optimizer on $h'$ to maximize $\max_i \epsilon_{\mathcal{S}_i}(h' \circ g) - \min_i \epsilon_{\mathcal{S}_i}(h' \circ g)$ fixing $g$

        Project $\theta'$ onto the Euclidean ball $\|\theta' - \theta\| \leq \delta$

    Fixing $h'$, run the descent optimizer on $g, h$ to minimize

        error $= \frac{1}{n} \sum_i \epsilon_{\mathcal{S}_i}(h \circ g) + (\max_i \epsilon_{\mathcal{S}_i}(h' \circ g) - \min_i \epsilon_{\mathcal{S}_i}(h' \circ g))$

**Output:** feature embedding $g$, classifier $h$

---

## 5 Experiments

Gulrajani and Lopez-Paz [20] did extensive experiments on comparing DG algorithms, using the same neural architecture and data split. Specifically, they show that with data augmentation, ERM perform relatively well among a large array of algorithms. Our experiments are based on their settings. We run Algorithm 1 on standard benchmarks, including RotatedMNIST [18], PACS [28], Office-Home [51] and WILDS-FMoW [23] (c.f. Appendix C.1). Specifically, WILDS-FMoW is a large dataset with nearly half a million images. Detailed experimental settings can be seen at Appendix C.

**Evaluating transferability** From Figure 3 it can be seen that at a neighborhood of the learned classifier, there exists a classifier such that the target accuracy is degraded significantly, whereas some source domain still has high accuracy. This poses questions to whether current popular algorithms such as ERM [50], DANN [17] and Mixup [53, 54] are really learning invariant and transferable features. If so, the target accuracy should be high given a high source accuracy. However, for the PACS dataset and Mixup model (the second column of Figure 3), the target accuracy decreases by more than $30\%$ while the source accuracy remains roughly at the same level. We can also, e.g., read from the first column that with a small decrease of the source (test) accuracy by $\sim 2\%$ (at $\delta = 2$), the target accuracy of DANN drops by $\sim 10\%$.

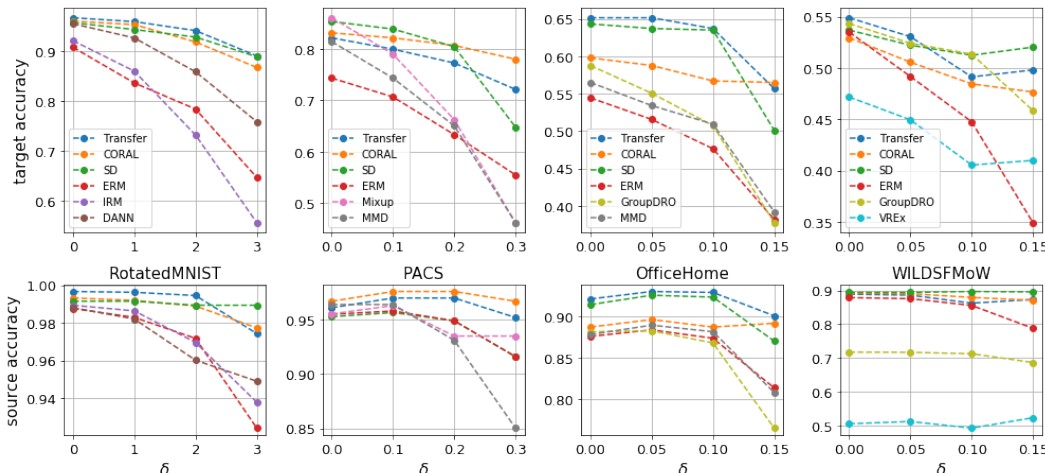

Figure 3: **Top row**: test accuracy of the target domain; **bottom row**: test accuracy of one of the source domains. Each column is for a given dataset with the name in the middle, and the legends on the bottom row are the same as those on the top row. $\delta$ is the parameter in Algorithm 1.

From Figure 3 we can also see that Correlation Alignment [CORAL, 47] and Spectral Decomposition [SD, 40] have better transferability that other algorithms. In some sense, they are in fact learning *robust classifiers*, i.e., all the classifiers on the neighborhood of the learned classifier can achieve good accuracies. With this robust classifier, the target accuracy does not decrease much even if the classifier is perturbed.

**Improving transferability** Algorithm 2 has good performance among all four datasets that we tried, comparable to CORAL and SD. Note that CORAL and SD do not always perform well, such as in the Office-Home and WILDS-FMoW datasets, but our Transfer algorithm does. However, in our experiments we find there are two limitations of Algorithm 2: **(1)** we need a large number of inner maximization steps to compute the gap, which needs more training time. This is similar to adversarial robustness [33] which is slower than usual training. In order to overcome this difficulty we used pretraining from other algorithms in the experiments on Office-Home and WILDS-FMoW; **(2)** Moderate hyper-parameter tuning is needed. For example, we need to tune $N$ is Algorithm 2, the learning rate (`lr`) of SGA and the choice of $\delta$. We find that taking $N = 20$ or $30$ is usually a good choice, and $\delta$ can be quite large such that the projection step is not taken. We take `lr` $= 0.01$ for RotatedMNIST and `lr` $= 0.001$ for other datasets.

**Label shift** In order to show the difference with the well-known $\mathcal{H}$-divergence [8], we compute the label shifts in the PACS dataset. As shown in Zhao et al. [59], the optimal joint error $\epsilon_{\mathcal{S}}(h^*) + \epsilon_{\mathcal{S}}(h^*)$ ($h^*$ is the optimal classifier that minimizes $\epsilon_{\mathcal{S}}(h) + \epsilon_{\mathcal{T}}(h)$) can be large under the shift of label distributions. We follow [59] and compute the label shift between pairs of domains in the PACS dataset, measured by total variation. From Table 1 we can see that the label shift is large in this case, and thus the $\mathcal{H}$-divergence bound [8] can be quite loose. Comparably, our transfer measure bound Prop. 3 is tighter (c.f. Prop. 27) and therefore still useful in practice.

Table 1: Label shift between pairs of domains in the PACS dataset. **TV:** total variation; **A:** art painting; **C:** cartoon; **P:** photo; **S:** sketch. The total variation is always between zero and one.

| TV | A | C | P | S |
|---|---|---|---|---|
| **A** | 0.0 | 0.12 | 0.11 | 0.3 |
| **C** | 0.12 | 0.0 | 0.18 | 0.24 |
| **P** | 0.11 | 0.18 | 0.0 | 0.37 |
| **S** | 0.3 | 0.24 | 0.37 | 0.0 |

# 6  Related Work

**Multi-task learning**  Multi-task learning (MTL) [56] is related to but different from DG. In MTL, there are several tasks, and one hopes to improve the performance of each task by jointly training all the tasks simultaneously, utilizing the relationships between them. This is different from DG in the sense that in DG the target domain is unknown a priori, whereas in MTL the focus is more on better generalization on existing tasks that appear in training. Hence, there is no distribution shift in MTL per se. Furthermore, for MTL, the output spaces of different tasks are not necessarily the same.

**Zero-shot learning / Few-shot learning / Meta-learning**  DG is different from zero-shot learning [27]. In zero-shot learning, one has labeled training data and the goal is to make predictions on a new unseen label set. However, in DG the label set remains the same for the source and the target domains. On the other hand, the focus of few-shot learning is on fast adaptation, in the sense that the test distribution remains the same as the training distribution, but the learner can only have access to a few labeled samples. Domain generalization also shares similarity with meta-learning. However, in meta-learning, the learner is allowed to fine-tune over the target domain. In other words, the protocol of meta-learning allows access to a small amount of labeled data from future unseen domains. Meta-learning is more or less one specific method that is used to tackle few-shot learning. Because of the similarity, some meta-learning algorithms can be applied to DG [29].

**Self-supervised learning**  Self-supervised learning (SSL) is a popular unsupervised feature learning approach [13, 19, 21]. The goal of SSL is to learn invariant representations w.r.t. different views of the same image. Although it is a promising feature learning method, it differs from our DG settings in the sense that no labels are used in SSL.

**Domain generalization**  There have been a lot of old and new algorithms proposed for domain generalization. The simplest one is Empirical Risk Minimization (ERM), where we simply minimize the empirical risk of (the sum of) all source domains. In Blanchard et al. [10], Muandet et al. [35], kernel methods for DG were proposed. Arjovsky et al. [3] proposed Invariant Risk Minimization (IRM) which aims to learn invariant predictors across source domains, and follow-up discussions can be found in [25, 41]. Another approach is called distributional robustness [43, 52], where the model is optimized over a worst-case distribution under the constraint that this distribution is generated from a small perturbation around the source distributions. In Albuquerque et al. [2], a DG scheme based on distribution matching was proposed. Moreover, many domain adaptation algorithms can be directly adapted to the task of domain generalization, such as CORAL [47] and DANN [16]. Last but not least, we mention a concurrent work [55] on the theory of domain generalization, which focuses more on proposing a model selection rule based on accuracy and variation.

**Adversarial robustness**  Our evaluation and training methods in §4 are reminiscent of the adversarial training method [33] in the literature of adversarial robustness. Perturbing the classifier in our case corresponds to perturbing the input data in adversarial robustness. From this perspective, our Transfer algorithm is parallel to the adversarial training method. It would be interesting to design certified robust feature embeddings, by analogy with certified robust classifiers [15].

# 7  Conclusions

In this paper we formally define the notion of *transferability* that we can quantify, estimate and compute. Our transfer measures can be understood as a special class of IPMs. They are weaker than total variation and even very dissimilar distributions could be transferable to each other. Our definition of transferability can also be naturally used to derive a generalization bound for prediction error on the target domain. Based on our theory, we propose algorithms to evaluate and improve the transferability by learning feature representations. Experiments show that, somewhat surprisingly, many existing algorithms are not quite learning transferable features. From this perspective, our transfer measures offer a novel way to evaluate the features learned from different DG algorithms. We hope that our proposal of transferability could draw the community's attention to further investigate and better understand the fundamental quantity that allows robust models under distribution shifts.

**Broader Impact**  Reliable domain generalization models are important for practice use. Our work points out the reliability issue of DG algorithms. It is worth mentioning that our evaluation method can only disprove the transferability and survival of our attack method should not be treated as a warranty. Misunderstanding of it could lead to potential harm in practical applications.

## Acknowledgements and Funding Transparency Statement

We thank the anonymous reviewers for their constructive comments as well as the area chair and the senior area chair for overseeing the review process. Resources used in preparing this research were provided, in part, by the Province of Ontario, the Government of Canada through CIFAR, and companies sponsoring the Vector Institute. We thank NSERC and the Canada CIFAR AI Chairs program for funding support. GZ is also supported by David R. Cheriton scholarship and research grant from Vector Institute. HZ is supported by a startup funding from the Department of Computer Science at UIUC. Finally, we thank Vector Institute for providing the GPU cluster.

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
