# A   Proofs

In this appendix, we present proofs of our theoretical results in the main paper.

**Proposition 4.** *For $\Gamma \subseteq \mathcal{H}$ and domains $\mathcal{S}, \mathcal{T}$ we have:* $\mathrm{T}_\Gamma(\mathcal{S}, \mathcal{T}) \leq 2\mathrm{T}_\Gamma^{\mathrm{r}}(\mathcal{S}, \mathcal{T})$.

*Proof.* We first note that:

$$\mathrm{T}_\Gamma(\mathcal{S}, \mathcal{T}) = \sup_{h \in \Gamma} |(\epsilon_\mathcal{S}(h) - \epsilon_\mathcal{T}(h) - (\epsilon_\mathcal{S}^* - \epsilon_\mathcal{T}^*))| \leq \sup_{h \in \Gamma} |\epsilon_\mathcal{S}(h) - \epsilon_\mathcal{T}(h)| + |\epsilon_\mathcal{S}^* - \epsilon_\mathcal{T}^*|$$
$$= \mathrm{T}_\Gamma^{\mathrm{r}}(\mathcal{S}, \mathcal{T}) + |\epsilon_\mathcal{S}^* - \epsilon_\mathcal{T}^*|. \tag{A.1}$$

It suffices to prove that $|\epsilon_\mathcal{S}^* - \epsilon_\mathcal{T}^*| \leq \mathrm{T}_\Gamma^{\mathrm{r}}(\mathcal{S}, \mathcal{T})$. Suppose $\mathrm{T}_\Gamma^{\mathrm{r}}(\mathcal{S}, \mathcal{T}) \leq \delta$, and $h_\mathcal{S}^* \in \mathrm{argmin}_{h \in \Gamma} \epsilon_\mathcal{S}$. Then $\epsilon_\mathcal{S}(h_\mathcal{S}^*) = \inf_{h \in \Gamma} \epsilon_\mathcal{S} = \epsilon_\mathcal{S}^*$ and we have:

$$\epsilon_\mathcal{T}^* - \delta \leq \epsilon_\mathcal{T}(h_\mathcal{S}^*) - \delta \leq \epsilon_\mathcal{S}(h_\mathcal{S}^*) = \epsilon_\mathcal{S}^*, \tag{A.2}$$

where in the first inequality, we used $\epsilon_\mathcal{T}^* = \inf_{h \in \Gamma} \epsilon_\mathcal{T}(h)$, and in the second inequality, we used that for any $h \in \Gamma$, we have:

$$\epsilon_\mathcal{T}(h) - \epsilon_\mathcal{S}(h) \leq \sup_{h \in \Gamma} |\epsilon_\mathcal{T}(h) - \epsilon_\mathcal{S}(h)| = \mathrm{T}_\Gamma^{\mathrm{r}}(\mathcal{S}, \mathcal{T}) \leq \delta. \tag{A.3}$$

Hence, we have $\epsilon_\mathcal{T}^* - \epsilon_\mathcal{S}^* \leq \mathrm{T}_\Gamma^{\mathrm{r}}(\mathcal{S}, \mathcal{T})$. Since $\mathrm{T}_\Gamma^{\mathrm{r}}(\mathcal{S}, \mathcal{T})$ is symmetric in $\mathcal{T}$ and $\mathcal{S}$, we also have $\epsilon_\mathcal{S}^* - \epsilon_\mathcal{T}^* \leq \mathrm{T}_\Gamma^{\mathrm{r}}(\mathcal{S}, \mathcal{T})$. Hence we have proved $|\epsilon_\mathcal{S}^* - \epsilon_\mathcal{T}^*| \leq \mathrm{T}_\Gamma^{\mathrm{r}}(\mathcal{S}, \mathcal{T})$. $\square$

**Proposition 5 (equivalence between transferability and transfer measures).** *Let $\delta_\mathcal{S} > 0$ and $\Gamma = \mathrm{argmin}(\epsilon_\mathcal{S}, \delta_\mathcal{S})$ and suppose $\inf_{h \in \Gamma} \epsilon_\mathcal{T}(h) = \inf_{h \in \mathcal{H}} \epsilon_\mathcal{T}(h)$. If $\mathrm{T}_\Gamma(\mathcal{S} \| \mathcal{T}) \leq \delta$ or $\mathrm{T}_\Gamma(\mathcal{S}, \mathcal{T}) \leq \delta$, then $\mathcal{S}$ is $(\delta_\mathcal{S}, \delta + \delta_\mathcal{S})$-transferable to $\mathcal{T}$. If $\mathcal{S}$ is $(\delta_\mathcal{S}, \delta_\mathcal{T})$-transferable to $\mathcal{T}$, then $\mathrm{T}_\Gamma(\mathcal{S} \| \mathcal{T}) \leq \delta_\mathcal{T}$ and $\mathrm{T}_\Gamma(\mathcal{S}, \mathcal{T}) \leq \max\{\delta_\mathcal{S}, \delta_\mathcal{T}\}$.*

*Proof.* If $\mathrm{T}_\Gamma(\mathcal{S} \| \mathcal{T}) = \sup_{h \in \Gamma} \epsilon_\mathcal{T}(h) - \epsilon_\mathcal{T}^* - (\epsilon_\mathcal{S}(h) - \epsilon_\mathcal{S}^*) \leq \delta$, then for any $h \in \Gamma = \mathrm{argmin}(\epsilon_\mathcal{S}, \delta_\mathcal{S})$ we have $\epsilon_\mathcal{S}(h) \leq \epsilon_\mathcal{S}^* + \delta_\mathcal{S}$ and

$$\epsilon_\mathcal{T}(h) - \epsilon_\mathcal{T}^* \leq \epsilon_\mathcal{S}(h) - \epsilon_\mathcal{S}^* + \delta \leq \delta_\mathcal{S} + \delta. \tag{A.4}$$

Since we assume $\epsilon_\mathcal{T}^* = \inf_{h \in \Gamma} \epsilon_\mathcal{T}(h) = \inf_{h \in \mathcal{H}} \epsilon_\mathcal{T}(h)$, we obtain that $\mathcal{S}$ is $(\delta_\mathcal{S}, \delta_\mathcal{S} + \delta)$-transferable to $\mathcal{T}$. If $\mathrm{T}_\Gamma(\mathcal{S}, \mathcal{T}) \leq \delta$, then $\mathrm{T}_\Gamma(\mathcal{S} \| \mathcal{T}) \leq \delta$, and $\mathcal{S}$ is $(\delta_\mathcal{S}, \delta_\mathcal{S} + \delta)$-transferable to $\mathcal{T}$.

If $\mathcal{S}$ is $(\delta_\mathcal{S}, \delta_\mathcal{T})$-transferable to $\mathcal{T}$, then for any $h \in \mathrm{argmin}(\epsilon_\mathcal{S}, \delta_\mathcal{S})$, we have:

$$\epsilon_\mathcal{T}(h) - \epsilon_\mathcal{T}^* - (\epsilon_\mathcal{S}(h) - \epsilon_\mathcal{S}^*) \leq \epsilon_\mathcal{T}(h) - \epsilon_\mathcal{T}^* \leq \delta_\mathcal{T}. \tag{A.5}$$

We also have $\mathrm{T}_\Gamma(\mathcal{S} \| \mathcal{T}) \leq \delta_\mathcal{T}$ from (A.5). Moreover, we can derive that:

$$\mathrm{T}_\Gamma(\mathcal{T} \| \mathcal{S}) = \sup_{h \in \Gamma} (\epsilon_\mathcal{S}(h) - \epsilon_\mathcal{S}^*) - (\epsilon_\mathcal{T}(h) - \epsilon_\mathcal{T}^*) \leq \sup_{h \in \Gamma} (\epsilon_\mathcal{S}(h) - \epsilon_\mathcal{S}^*) \leq \delta_\mathcal{S}, \tag{A.6}$$

and thus $\mathrm{T}_\Gamma(\mathcal{S}, \mathcal{T}) \leq \max\{\delta_\mathcal{S}, \delta_\mathcal{T}\}$ from the definition. $\square$

**Proposition 6 (pseudo-metric).** *For a general loss $\epsilon_\mathcal{D}$ as in (1), $\mathrm{T}_\Gamma^{\mathrm{r}}(\mathcal{S}, \mathcal{T})$ is a pseudo-metric, i.e., for any distributions $\mathcal{S}, \mathcal{T}, \mathcal{P}$ on the same underlying space, we have $\mathrm{T}_\Gamma^{\mathrm{r}}(\mathcal{S}, \mathcal{S}) = 0$, $\mathrm{T}_\Gamma^{\mathrm{r}}(\mathcal{S}, \mathcal{T}) = \mathrm{T}_\Gamma^{\mathrm{r}}(\mathcal{T}, \mathcal{S})$ (symmetry), and $\mathrm{T}_\Gamma^{\mathrm{r}}(\mathcal{S}, \mathcal{T}) \leq \mathrm{T}_\Gamma^{\mathrm{r}}(\mathcal{S}, \mathcal{P}) + \mathrm{T}_\Gamma^{\mathrm{r}}(\mathcal{P}, \mathcal{T})$ (triangle inequality).*

*Proof.* $\mathrm{T}_\Gamma(\mathcal{S}, \mathcal{S}) = 0$ and $\mathrm{T}_\Gamma(\mathcal{S}, \mathcal{T}) = \mathrm{T}_\Gamma(\mathcal{T}, \mathcal{S})$ follow from the definition. Denote the excess risk $\mathrm{exc}_\mathcal{S}(h) = \epsilon_\mathcal{S}(h) - \epsilon_\mathcal{S}^*$ (we could change the letter $\mathcal{S}$ here). The triangle inequality can be derived as:

$$\mathrm{T}_\Gamma(\mathcal{S}, \mathcal{T}) = \sup_{h \in \Gamma} |\mathrm{exc}_\mathcal{S}(h) - \mathrm{exc}_\mathcal{T}(h)|$$
$$= \sup_{h \in \Gamma} |\mathrm{exc}_\mathcal{S}(h) - \mathrm{exc}_\mathcal{P}(h) + \mathrm{exc}_\mathcal{P}(h) - \mathrm{exc}_\mathcal{T}(h)|$$
$$\leq \sup_{h \in \Gamma} |\mathrm{exc}_\mathcal{S}(h) - \mathrm{exc}_\mathcal{P}(h)| + \sup_{h \in \Gamma} |\mathrm{exc}_\mathcal{P}(h) - \mathrm{exc}_\mathcal{T}(h)|$$
$$= \mathrm{T}_\Gamma(\mathcal{S}, \mathcal{P}) + \mathrm{T}_\Gamma(\mathcal{P}, \mathcal{T}). \tag{A.7}$$

Similarly, we can derive $\mathrm{T}_\Gamma(\mathcal{S} \| \mathcal{T}) \leq \mathrm{T}_\Gamma(\mathcal{S} \| \mathcal{P}) + \mathrm{T}_\Gamma(\mathcal{P} \| \mathcal{T})$. $\square$

**Proposition 7** (**equivalence with total variation**). *For binary classification with labels $\{-1, 1\}$, given the 0-1 loss $\epsilon_{\mathcal{D}} = \epsilon_{\mathcal{D}}^{0-1}$, we have $\mathrm{T}_{\Gamma}^{\mathrm{r}}(\mathcal{S}, \mathcal{T}) \leq d_{\mathrm{TV}}(\mathcal{S}, \mathcal{T})$ for domains $\mathcal{S}, \mathcal{T}$ and any $\Gamma \subseteq \mathcal{H}$. Denote $\mathcal{H}_t$ to be the set of all binary classifiers. Then we have $d_{\mathrm{TV}}(\mathcal{S}, \mathcal{T}) \leq 4\mathrm{T}_{\mathcal{H}_t}^{\mathrm{r}}(\mathcal{S}, \mathcal{T})$.*

*Proof.* Let us first recall the definition of IPMs:

$$d_{\mathcal{F}}(\mathcal{S}, \mathcal{T}) = \sup_{f \in \mathcal{F}} \left| \sum_y \int f(x, y)(p_{\mathcal{S}}(x, y) - p_{\mathcal{T}}(x, y))dx \right|. \tag{A.8}$$

The symmetric transfer measure $\mathrm{T}_{\Gamma}^{\mathrm{r}}(\mathcal{S}, \mathcal{T})$ and the total variation can be represented as:

$$\mathrm{T}_{\Gamma}^{\mathrm{r}}(\mathcal{S}, \mathcal{T}) = d_{\mathcal{F}_{\Gamma}}(\mathcal{S}, \mathcal{T}), \ d_{\mathrm{TV}}(\mathcal{S}, \mathcal{T}) = d_{\mathcal{F}_{\mathrm{TV}}}(\mathcal{S}, \mathcal{T}), \tag{A.9}$$

with $\mathcal{F}_{\Gamma} := \{(x, y) \mapsto \mathbb{1}(h(x) \neq y), h \in \Gamma\}$, $\mathcal{F}_{\mathrm{TV}} = \{f : \|f\|_{\infty} \leq 1\}$ (see also Appendix B.2). The first sentence follows from $\mathcal{F}_{\Gamma} \subseteq \mathcal{F}_{\mathrm{TV}}$, and the definition of IPM.

Now let us prove the case when $\Gamma = \mathcal{H}_t$ is unconstrained. Suppose $\mathrm{T}_{\mathcal{H}_t}^{\mathrm{r}}(\mathcal{S}, \mathcal{T}) \leq \delta$, then for any binary classifier $h$, we have $|\epsilon_{\mathcal{S}}(h) - \epsilon_{\mathcal{T}}(h)| \leq \delta$. For simplicity, denote the difference of the two distributions as:

$$d(x, y) := p_{\mathcal{S}}(x, y) - p_{\mathcal{T}}(x, y). \tag{A.10}$$

Take $h_+$ to be the following (note that we allow the classifier to take a garbage value 0):

$$h_+(x) = \begin{cases} 0 & \text{if } x \in \mathcal{B}_{>>} := \{x \in \mathcal{X} : d(x, 1) \geq 0 \text{ and } d(x, -1) \geq 0\} \\ -1 & \text{if } x \in \mathcal{B}_{><} := \{x \in \mathcal{X} : d(x, 1) \geq 0, \ d(x, -1) < 0\} \\ 1 & \text{if } x \in \mathcal{B}_{<>} := \{x \in \mathcal{X} : d(x, 1) < 0, \ d(x, -1) \geq 0\} \\ 1 & \text{if } x \in \mathcal{B}_{<<}^- := \{x \in \mathcal{X} : d(x, 1) < d(x, -1) < 0\} \\ -1 & \text{if } x \in \mathcal{B}_{<<}^+ := \{x \in \mathcal{X} : 0 > d(x, 1) \geq d(x, -1)\} \end{cases}, \tag{A.11}$$

and denote $\mathcal{B}_{<<} := \mathcal{B}_{<<}^- \cup \mathcal{B}_{<<}^+$. Then we have from the definition:

$$\epsilon_{\mathcal{S}}(h_+) - \epsilon_{\mathcal{T}}(h_+) = \sum_y \int (p_{\mathcal{S}}(x, y) - p_{\mathcal{T}}(x, y))\mathbb{1}(h_+(x) \neq y)dx$$

$$= \int d(x, 1)\mathbb{1}(h_+(x) \neq 1) + d(x, -1)\mathbb{1}(h_+(x) \neq -1)dx$$

$$= \int_{\mathcal{B}_{>>}} d(x, 1) + d(x, -1)dx + \int_{\mathcal{B}_{><}} d(x, 1)dx + \int_{\mathcal{B}_{<>}} d(x, -1)dx$$

$$- \int_{\mathcal{B}_{<<}} \min\{-d(x, 1), -d(x, -1)\}dx. \tag{A.12}$$

Moreover, one can verify that $\epsilon_{\mathcal{S}}(h_+) - \epsilon_{\mathcal{T}}(h_+) = \sup_{h \in \mathcal{H}_t} \epsilon_{\mathcal{S}}(h) - \epsilon_{\mathcal{T}}(h)$. Similarly, let us define $h_-$ to be:

$$h_-(x) = \begin{cases} 0 & \text{if } x \in \mathcal{B}_{<<} := \{x \in \mathcal{X} : d(x, 1) < 0 \text{ and } d(x, -1) < 0\} \\ -1 & \text{if } x \in \mathcal{B}_{<>} := \{x \in \mathcal{X} : d(x, 1) < 0, \ d(x, -1) \geq 0\} \\ 1 & \text{if } x \in \mathcal{B}_{><} := \{x \in \mathcal{X} : d(x, 1) \geq 0, \ d(x, -1) < 0\} \\ -1 & \text{if } x \in \mathcal{B}_{>>}^- := \{x \in \mathcal{X} : 0 \leq d(x, 1) < d(x, -1)\} \\ 1 & \text{if } x \in \mathcal{B}_{>>}^+ := \{x \in \mathcal{X} : d(x, 1) \geq d(x, -1) \geq 0\} \end{cases}. \tag{A.13}$$

Then we have from the definition:

$$\epsilon_{\mathcal{T}}(h_-) - \epsilon_{\mathcal{S}}(h_-) = -\sum_y \int (p_{\mathcal{S}}(x, y) - p_{\mathcal{T}}(x, y))\mathbb{1}(h_-(x) \neq y)dx$$

$$= \int -d(x, 1)\mathbb{1}(h_-(x) \neq 1) - d(x, -1)\mathbb{1}(h_-(x) \neq -1)dx$$

$$= \int_{\mathcal{B}_{<<}} -d(x, 1) - d(x, -1)dx + \int_{\mathcal{B}_{><}} -d(x, -1)dx + \int_{\mathcal{B}_{<>}} -d(x, 1)dx$$

$$- \int_{\mathcal{B}_{>>}} \min\{d(x, 1), d(x, -1)\}dx. \tag{A.14}$$

Moreover, $\epsilon_{\mathcal{T}}(h_-) - \epsilon_{\mathcal{S}}(h_-) = \sup_{h \in \mathcal{H}_t} \epsilon_{\mathcal{T}}(h) - \epsilon_{\mathcal{S}}(h)$. Summing over (A.12) and (A.14) we have:

$$2 \sup_{h \in \mathcal{H}_t} |\epsilon_{\mathcal{S}}(h) - \epsilon_{\mathcal{T}}(h)| \geq |\epsilon_{\mathcal{S}}(h_+) - \epsilon_{\mathcal{T}}(h_+)| + |\epsilon_{\mathcal{T}}(h_-) - \epsilon_{\mathcal{S}}(h_-)|$$

$$\geq \epsilon_{\mathcal{S}}(h_+) - \epsilon_{\mathcal{T}}(h_+) + \epsilon_{\mathcal{T}}(h_-) - \epsilon_{\mathcal{S}}(h_-)$$

$$= \int_{\mathcal{B}_{>>}} \max\{d(x,1), d(x,-1)\}dx + \int_{\mathcal{B}_{><}} d(x,1) - d(x,-1)dx$$

$$+ \int_{\mathcal{B}_{<>}} -d(x,1) + d(x,-1)dx + \int_{\mathcal{B}_{<<}} \max\{-d(x,1), -d(x,-1)\}dx.$$

$$(A.15)$$

On the other hand, we can compute the total variation between $\mathcal{S}$ and $\mathcal{T}$:

$$d_{\mathrm{TV}}(\mathcal{S}, \mathcal{T}) = \sum_y \int |p_{\mathcal{S}}(x,y) - p_{\mathcal{T}}(x,y)|dx$$

$$= \int |d(x,1)| + |d(x,-1)|dx$$

$$= \int_{\mathcal{B}_{>>}} d(x,1) + d(x,-1)dx + \int_{\mathcal{B}_{><}} d(x,1) - d(x,-1)dx$$

$$+ \int_{\mathcal{B}_{<>}} -d(x,1) + d(x,-1)dx + \int_{\mathcal{B}_{<<}} -d(x,1) - d(x,-1)dx$$

$$\leq 2 \int_{\mathcal{B}_{>>}} \max\{d(x,1), d(x,-1)\}dx + 2 \int_{\mathcal{B}_{><}} d(x,1) - d(x,-1)dx$$

$$+ 2 \int_{\mathcal{B}_{<>}} -d(x,1) + d(x,-1)dx + \int_{\mathcal{B}_{<<}} 2 \max\{-d(x,1), -d(x,-1)\}dx$$

$$\leq 4 \sup_{h \in \mathcal{H}_t} |\epsilon_{\mathcal{S}}(h) - \epsilon_{\mathcal{T}}(h)| = 4 \mathrm{T}^{\mathrm{r}}_{\mathcal{H}_t}(\mathcal{S}, \mathcal{T}),$$

$$(A.16)$$

where in the last line we used (A.15). $\qquad\square$

In the proof above, we assumed a classifier $h \in \Gamma$ is allowed to take a garbage value 0 if it is not sure which label to choose. This is a mild assumption that can hold in practice.

**Lemma 9'** (**reduction of estimation error**). *Suppose $\widehat{\mathcal{S}}$ and $\widehat{\mathcal{T}}$ are i.i.d. sample distributions drawn from distributions of $\mathcal{S}$ and $\mathcal{T}$, then for any $\Gamma \subseteq \mathcal{H}$ we have:*

$$\mathrm{T}_\Gamma(\mathcal{S}\|\mathcal{T}) \leq \mathrm{T}_\Gamma(\widehat{\mathcal{S}}\|\widehat{\mathcal{T}}) + 2\mathrm{est}_\Gamma(\mathcal{S}) + 2\mathrm{est}_\Gamma(\mathcal{T}), \tag{A.17}$$

$$\mathrm{T}_\Gamma(\mathcal{S}, \mathcal{T}) \leq \mathrm{T}_\Gamma(\widehat{\mathcal{S}}, \widehat{\mathcal{T}}) + 2\mathrm{est}_\Gamma(\mathcal{S}) + 2\mathrm{est}_\Gamma(\mathcal{T}), \tag{A.18}$$

$$\mathrm{T}^{\mathrm{r}}_\Gamma(\mathcal{S}, \mathcal{T}) \leq \mathrm{T}^{\mathrm{r}}_\Gamma(\widehat{\mathcal{S}}, \widehat{\mathcal{T}}) + \mathrm{est}_\Gamma(\mathcal{S}) + \mathrm{est}_\Gamma(\mathcal{T}), \tag{A.19}$$

*where we define*

$$\mathrm{est}_\Gamma(\mathcal{S}) = \sup_{h \in \Gamma} |\epsilon_{\mathcal{S}}(h) - \epsilon_{\widehat{\mathcal{S}}}(h)|, \ \mathrm{est}_\Gamma(\mathcal{T}) = \sup_{h \in \Gamma} |\epsilon_{\mathcal{T}}(h) - \epsilon_{\widehat{\mathcal{T}}}(h)|. \tag{A.20}$$

*Proof.* We prove the first inequality for example and others follow similarly. Note that:

$$\epsilon_{\mathcal{T}}(h) - \epsilon^*_{\mathcal{T}} - \epsilon_{\mathcal{S}}(h) + \epsilon^*_{\mathcal{S}} = \epsilon_{\mathcal{T}}(h) - \epsilon_{\widehat{\mathcal{T}}}(h) + \epsilon_{\widehat{\mathcal{T}}}(h) - \epsilon^*_{\mathcal{T}} - \epsilon^*_{\widehat{\mathcal{T}}} + \epsilon^*_{\widehat{\mathcal{T}}} - \epsilon_{\mathcal{S}}(h) - \epsilon_{\widehat{\mathcal{S}}}(h) + \epsilon_{\widehat{\mathcal{S}}}(h)+$$

$$+ \epsilon^*_{\mathcal{S}} - \epsilon^*_{\widehat{\mathcal{S}}} + \epsilon^*_{\widehat{\mathcal{S}}}$$

$$= (\epsilon_{\widehat{\mathcal{T}}}(h) - \epsilon^*_{\widehat{\mathcal{T}}} - \epsilon_{\widehat{\mathcal{S}}}(h) + \epsilon^*_{\widehat{\mathcal{S}}}) + (\epsilon_{\mathcal{T}}(h) - \epsilon_{\widehat{\mathcal{T}}}(h)) + (\epsilon^*_{\widehat{\mathcal{T}}} - \epsilon^*_{\mathcal{T}})+$$

$$+ (\epsilon_{\widehat{\mathcal{S}}}(h) - \epsilon_{\mathcal{S}}(h)) + (\epsilon^*_{\mathcal{S}} - \epsilon^*_{\widehat{\mathcal{S}}}). \tag{A.21}$$

Taking the supremum on both sides we have:

$$\mathrm{T}_\Gamma(\mathcal{S}\|\mathcal{T}) \leq \mathrm{T}_\Gamma(\widehat{\mathcal{S}}\|\widehat{\mathcal{T}}) + \sup_{h \in \Gamma} |\epsilon_{\mathcal{T}}(h) - \epsilon_{\widehat{\mathcal{T}}}(h)| + \epsilon^*_{\widehat{\mathcal{T}}} - \epsilon^*_{\mathcal{T}} + \sup_{h \in \Gamma} |\epsilon_{\widehat{\mathcal{S}}}(h) - \epsilon_{\mathcal{S}}(h)| + \epsilon^*_{\mathcal{S}} - \epsilon^*_{\widehat{\mathcal{S}}}.$$

$$(A.22)$$

Take $h_{\mathcal{T}}^* \in \operatorname{argmin}_{h \in \Gamma} \epsilon_{\mathcal{T}}(h)$ to be an optimal classifier. We can derive:

$$\epsilon_{\widehat{\mathcal{T}}}^* \leq \epsilon_{\widehat{\mathcal{T}}}(h_{\mathcal{T}}^*) \leq \epsilon_{\mathcal{T}}^*(h_{\mathcal{T}}^*) + \operatorname{est}_{\Gamma}(\mathcal{T}) = \epsilon_{\mathcal{T}}^* + \operatorname{est}_{\Gamma}(\mathcal{T}). \tag{A.23}$$

Therefore, $\epsilon_{\widehat{\mathcal{T}}}^* - \epsilon_{\mathcal{T}}^* \leq \operatorname{est}_{\Gamma}(\mathcal{T})$. Similarly, $\epsilon_{\mathcal{S}}^* - \epsilon_{\widehat{\mathcal{S}}}^* \leq \operatorname{est}_{\Gamma}(\mathcal{S})$. Combining all those above we obtain (A.17). $\qquad \square$

**Theorem 10' (estimation error with Rademacher complexity).** *Given 0-1 loss $\epsilon_{\mathcal{D}} = \epsilon_{\mathcal{D}}^{0-1}$, suppose $\widehat{\mathcal{S}}$ and $\widehat{\mathcal{T}}$ are sample sets with $m$ and $k$ samples drawn i.i.d. from distributions $\mathcal{S}$ and $\mathcal{T}$, respectively. For any $\Gamma \subseteq \mathcal{H}$ any of the following holds w.p. $1 - \delta$:*

$$\mathrm{T}_{\Gamma}(\mathcal{S}\|\mathcal{T}) \leq \mathrm{T}_{\Gamma}(\widehat{\mathcal{S}}\|\widehat{\mathcal{T}}) + 4\mathfrak{R}_m(\mathcal{F}_{\Gamma}) + 4\mathfrak{R}_k(\mathcal{F}_{\Gamma}) + 2\sqrt{\frac{\log(4/\delta)}{2m}} + 2\sqrt{\frac{\log(4/\delta)}{2k}}, \tag{A.24}$$

$$\mathrm{T}_{\Gamma}(\mathcal{S}, \mathcal{T}) \leq \mathrm{T}_{\Gamma}(\widehat{\mathcal{S}}, \widehat{\mathcal{T}}) + 4\mathfrak{R}_m(\mathcal{F}_{\Gamma}) + 4\mathfrak{R}_k(\mathcal{F}_{\Gamma}) + 2\sqrt{\frac{\log(4/\delta)}{2m}} + 2\sqrt{\frac{\log(4/\delta)}{2k}}, \tag{A.25}$$

$$\mathrm{T}_{\Gamma}^{\mathrm{r}}(\mathcal{S}, \mathcal{T}) \leq \mathrm{T}_{\Gamma}^{\mathrm{r}}(\widehat{\mathcal{S}}, \widehat{\mathcal{T}}) + 2\mathfrak{R}_m(\mathcal{F}_{\Gamma}) + 2\mathfrak{R}_k(\mathcal{F}_{\Gamma}) + \sqrt{\frac{\log(4/\delta)}{2m}} + \sqrt{\frac{\log(4/\delta)}{2k}}, \tag{A.26}$$

*where $\mathcal{F}_{\Gamma} := \{(z, y) \mapsto \mathbb{1}(h(z) \neq y), h \in \Gamma\}$. If furthermore, $\Gamma$ is a set of binary classifiers with labels $\{-1, 1\}$, then $2\mathfrak{R}_m(\mathcal{F}_{\Gamma}) = \mathfrak{R}_m(\Gamma)$, $2\mathfrak{R}_k(\mathcal{F}_{\Gamma}) = \mathfrak{R}_k(\Gamma)$.*

*Proof.* We use the following lemma, which a slight adaptation of Mohri et al. [34], Theorem 3.3:

**Lemma 11.** *Let $\mathcal{F}$ be a family of functions from $\mathcal{X} \times \mathcal{Y}$ to $[0, 1]$. Then for any $\delta > 0$, with probability at least $1 - \delta$ over the draw from a distribution $\mathcal{S}$ of an i.i.d. samples $S$ of size $m$, $\{w_i\}_{i=1}^m$, the following holds for all $f \in \mathcal{F}$,*

$$\left| \mathbb{E}[f(w)] - \frac{1}{m}\sum_{i=1}^m f(w_i) \right| \leq 2\mathfrak{R}_m(\mathcal{F}) + \sqrt{\frac{\log(2/\delta)}{2m}}. \tag{A.27}$$

*Proof.* From Mohri et al. [34], Theorem 3.3, we know with probability at least $1 - \delta/2$, the following holds

$$\mathbb{E}[f(w)] - \frac{1}{m}\sum_{i=1}^m f(w_i) \leq 2\mathfrak{R}_m(\mathcal{F}) + \sqrt{\frac{\log(2/\delta)}{2m}}. \tag{A.28}$$

This result relies on applying McDiarmid's inequality on $\Phi(S) = \sup_{f \in \mathcal{F}} \mathbb{E}[f] - \frac{1}{m}\sum_{i=1}^m f(w_i)$. By repeating the same proof and applying McDiarmid's inequality on $\Phi'(S) = \sup_{f \in \mathcal{F}} \frac{1}{m}\sum_{i=1}^m f(w_i) - \mathbb{E}[f]$, we conclude that with probability at least $1 - \delta/2$, the following holds

$$\frac{1}{m}\sum_{i=1}^m f(w_i) - \mathbb{E}[f(w)] \leq 2\mathfrak{R}_m(\mathcal{F}) + \sqrt{\frac{\log(2/\delta)}{2m}}. \tag{A.29}$$

Therefore, with union bound we obtain that with probability (w.p.) at least $1 - \delta$, we have (A.27).

$\qquad \square$

Let us now go back to the proof of Theorem 10'. Taking $\mathcal{F}_{\Gamma} = \{(z, y) \mapsto \mathbb{1}(h(z) \neq y), h \in \Gamma\}$, we can derive from the theorem above that w.p. at least $1 - \delta$:

$$\operatorname{est}_{\Gamma}(\mathcal{S}) = \sup_{h \in \Gamma} |\epsilon_{\mathcal{S}}(h) - \epsilon_{\widehat{\mathcal{S}}}(h)| \leq 2\mathfrak{R}_m(\mathcal{F}_{\Gamma}) + \sqrt{\frac{\log(2/\delta)}{2m}}. \tag{A.30}$$

With (A.30) we know that with probability at least $1 - \delta/2$:

$$\operatorname{est}_{\Gamma}(\mathcal{S}) = \sup_{h \in \Gamma} |\epsilon_{\mathcal{S}}(h) - \epsilon_{\widehat{\mathcal{S}}}(h)| \leq 2\mathfrak{R}_m(\mathcal{F}_{\Gamma}) + \sqrt{\frac{\log(4/\delta)}{2m}}, \tag{A.31}$$

and w.p. at least $1 - \delta/2$:

$$\text{est}_\Gamma(\mathcal{T}) = \sup_{h \in \Gamma} |\epsilon_\mathcal{T}(h) - \epsilon_{\hat{\mathcal{T}}}(h)| \leq 2\mathfrak{R}_k(\mathcal{F}_\Gamma) + \sqrt{\frac{\log(4/\delta)}{2k}}, \tag{A.32}$$

therefore from union bound w.p. at least $1 - \delta$ we have:

$$\text{est}_\Gamma(\mathcal{S}) + \text{est}_\Gamma(\mathcal{T}) \leq 2\mathfrak{R}_m(\mathcal{F}_\Gamma) + 2\mathfrak{R}_k(\mathcal{F}_\Gamma) + \sqrt{\frac{\log(4/\delta)}{2m}} + \sqrt{\frac{\log(4/\delta)}{2k}}. \tag{A.33}$$

Moreover, from Lemma 3.4 of Mohri et al. [34] we have

$$2\mathfrak{R}_m(\mathcal{F}_\Gamma) = \mathfrak{R}_m(\Gamma), \quad 2\mathfrak{R}_k(\mathcal{F}_\Gamma) = \mathfrak{R}_k(\Gamma), \tag{A.34}$$

for binary classification. The rest follows from Lemma 9'. $\square$

**Proposition 11 (transfer measure with a surrogate loss).** *Given surrogate loss $\epsilon_\mathcal{D} \geq \epsilon_\mathcal{D}^{0-1}$ on a general domain $\mathcal{D}$. Suppose $\Gamma = \text{argmin}(\epsilon_\mathcal{S}, \delta_\mathcal{S})$ and denote $\epsilon_\mathcal{T}^* = \inf_{h \in \Gamma} \epsilon_\mathcal{T}(h)$, $\epsilon_\mathcal{S}^* = \inf_{h \in \Gamma} \epsilon_\mathcal{S}(h)$, $(\epsilon_\mathcal{T}^{0-1})^* = \inf_{h \in \mathcal{H}} \epsilon_\mathcal{T}^{0-1}(h)$. If the following holds:*

$$\text{T}_\Gamma(\mathcal{S} \| \mathcal{T}) = \sup_{h \in \Gamma} \epsilon_\mathcal{T}(h) - \epsilon_\mathcal{T}^* - (\epsilon_\mathcal{S}(h) - \epsilon_\mathcal{S}^*) \leq \delta, \tag{8}$$

*then we have $\text{argmin}(\epsilon_\mathcal{S}, \delta_\mathcal{S}) \subseteq \text{argmin}(\epsilon_\mathcal{T}^{0-1}, \delta + \delta_\mathcal{S} + \epsilon_\mathcal{T}^* - (\epsilon_\mathcal{T}^{0-1})^*)$.*

*Proof.* Suppose (8) holds and thus for any $h \in \text{argmin}(\epsilon_\mathcal{S}, \delta_\mathcal{S})$ we have:

$$\epsilon_\mathcal{T}^{0-1}(h) \leq \epsilon_\mathcal{T}(h) \leq (\epsilon_\mathcal{S}(h) - \epsilon_\mathcal{S}^*) + \delta + \epsilon_\mathcal{T}^* \leq \delta_\mathcal{S} + \delta + \epsilon_\mathcal{T}^*. \tag{A.35}$$

The rest follows from definitions. $\square$

**Proposition 12 (domain generalization guarantee).** *Suppose we have $n$ distributions $\mathcal{S}_1^g, \ldots, \mathcal{S}_n^g$ which satisfy*

$$\sup_{h \in \Gamma} \max_i \epsilon_{\mathcal{S}_i^g}(h) - \min_j \epsilon_{\mathcal{S}_j^g}(h) \leq \delta. \tag{A.36}$$

*Then for any two distributions $\mathcal{T}_1^g, \mathcal{T}_2^g$ in $\text{conv}(\mathcal{S}_1^g, \ldots, \mathcal{S}_n^g)$, we have $\text{T}_\Gamma^r(\mathcal{T}_1^g, \mathcal{T}_2^g) \leq \delta$.*

*Proof.* For the ease of notation we omit the superscript $g$ in the proof. We treat distributions as probabilistic measures and thus for any $h \in \mathcal{H}$, $\epsilon_\mathcal{D}(h)$ is a linear function of $\mathcal{D}$, if we treat $\mathcal{D}$ as a probability measure. It suffices to prove for a linear function $f$, we have:

$$|f(\sum_i \pi_i \mathcal{S}_i) - f(\sum_j \pi_j' \mathcal{S}_j)| \leq \max_i f(\mathcal{S}_i) - \min_j f(\mathcal{S}_j), \tag{A.37}$$

where $\pi_i, \pi_j' \geq 0$ and $\sum_i \pi_i = \sum_j \pi_j' = 1$. This is because

$$|f(\sum_i \pi_i \mathcal{S}_i) - f(\sum_j \pi_j' \mathcal{S}_j)| = |f(\sum_{i,j} \pi_i \pi_j' \mathcal{S}_i) - f(\sum_{i,j} \pi_i \pi_j' \mathcal{S}_j))|$$

$$= |f(\sum_{i,j} \pi_i \pi_j' (\mathcal{S}_i - \mathcal{S}_j))|$$

$$= |\sum_{i,j} \pi_i \pi_j' f(\mathcal{S}_i - \mathcal{S}_j)|$$

$$\leq \sum_{i,j} \pi_i \pi_j' |f(\mathcal{S}_i - \mathcal{S}_j)|$$

$$\leq \max_{i,j} |f(\mathcal{S}_i) - f(\mathcal{S}_j)|$$

$$= \max_i f(\mathcal{S}_i) - \min_j f(\mathcal{S}_j). \tag{A.38}$$

The second and the third lines follow from the linearity of $f$ and the fourth line follows from triangle inequality. Therefore, taking $f : \mathcal{D} \mapsto \epsilon_\mathcal{D}(h)$ for any $h \in \Gamma$, and $\mathcal{T}_1 = \sum_i \pi_i \mathcal{S}_i$, $\mathcal{T}_2 = \sum_j \pi_j' \mathcal{S}_j$, we can derive from (A.38) that:

$$|\epsilon_{\mathcal{T}_1}(h) - \epsilon_{\mathcal{T}_2}(h)| \leq \max_i \epsilon_{\mathcal{S}_i}(h) - \min_j \epsilon_{\mathcal{S}_j}(h), \tag{A.39}$$

for any $h \in \Gamma$. Taking the supremum over $h$ on both sides we finish the proof. $\square$

**Theorem 12** (**optimization guarantee**). *Assume that the function $q(\cdot, x)$ is $L_\theta$ Lipschitz continuous for any $x$. Suppose we have learned a feature embedding $g$ and a classifier $h$ such that the loss functional $\epsilon_{\mathcal{S}_i^g} : \mathcal{H} \to \mathbb{R}$ is $L_\ell$ Lipschitz continuous w.r.t. distribution $\mathcal{S}_i^g$ for $i \in [n]$ and*

$$\max_{\|\theta' - \theta\| \le \delta} \frac{1}{n} \sum_{i=1}^{n} \epsilon_{\mathcal{S}_i}(h \circ g) + \left( \max_i \epsilon_{\mathcal{S}_i}(h' \circ g) - \min_i \epsilon_{\mathcal{S}_i}(h' \circ g) \right) \le \eta, \tag{15}$$

*where $\theta, \theta'$ are parameters of $h$ and $h'$. Then for any $h' \in \Gamma = \{q(\theta', \cdot) : \|\theta - \theta'\| \le \delta\}$, we have:*

$$\mathrm{T}_\Gamma^{\mathrm{r}}(\mathcal{T}_1^g, \mathcal{T}_2^g) \le \eta, \quad \epsilon_{\mathcal{S}_i}(h' \circ g) \le \eta + L_\ell L_\theta \delta, \quad \epsilon_{\mathcal{T}}(h' \circ g) \le 2\eta + L_\ell L_\theta \delta, \tag{16}$$

*for any $\mathcal{T}_1^g, \mathcal{T}_2^g, \mathcal{T}^g \in \mathrm{conv}(\mathcal{S}_1^g, \dots, \mathcal{S}_n^g)$ and any $i \in [n]$.*

*Proof.* From (15) we know that:

$$\max_i \epsilon_{\mathcal{S}_i}(h' \circ g) - \min_i \epsilon_{\mathcal{S}_i}(h' \circ g) \le \eta, \tag{A.40}$$

for any $h' = q(\theta', \cdot)$ and $\|\theta' - \theta\|_2 \le \delta$. Taking $h' = h$, we obtain that:

$$\begin{aligned}
\max_i \epsilon_{\mathcal{S}_i}(h \circ g) &= \min_i \epsilon_{\mathcal{S}_i}(h \circ g) + \max_i \epsilon_{\mathcal{S}_i}(h \circ g) - \min_i \epsilon_{\mathcal{S}_i}(h \circ g) \\
&\le \frac{1}{n} \sum_{i=1}^{n} \epsilon_{\mathcal{S}_i}(h \circ g) + \max_i \epsilon_{\mathcal{S}_i}(h \circ g) - \min_i \epsilon_{\mathcal{S}_i}(h \circ g) \\
&\le \eta. \tag{A.41}
\end{aligned}$$

In other words, for any $i \in [n] = \{1, \dots, n\}$, $\epsilon_{\mathcal{S}_i}(h \circ g) \le \eta$ holds. We have from Theorem 25 $\|h - h'\|_{1, \mathcal{D}} \le L_\theta \delta$ for any probability measure $\mathcal{D}$. Using Definition 19 we know that $|\epsilon_{\mathcal{S}_i}(h' \circ g) - \epsilon_{\mathcal{S}_i}(h \circ g)| \le L_\ell L_\theta \delta$. Therefore, for any $h' \in \Gamma$, we have:

$$\epsilon_{\mathcal{S}_i}(h' \circ g) \le \epsilon_{\mathcal{S}_i}(h \circ g) + L_\ell L_\theta \delta \le \eta + L_\ell L_\theta \delta. \tag{A.42}$$

From (A.40) and Prop. 12, for any $\mathcal{T} \in \mathrm{conv}(\mathcal{S}_1, \dots, \mathcal{S}_n)$ and any $\mathcal{S}_i$, $\mathrm{T}_\Gamma^{\mathrm{r}}(\mathcal{T}, \mathcal{S}_i) \le \eta$ holds, and thus from the definition of $\mathrm{T}_\Gamma^{\mathrm{r}}$ we have the third inequality of (16). The first inequality of (16) follows from Proposition 12. $\qquad\square$

# B   Additional theoretical results

In this appendix we present additional theoretical results as supplementary material.

## B.1   Necessity of excess risks

We give an example where the realizable transfer measure is large but the source domain is transferable to the target domain.

**Example 13.** *Consider two distributions:*

$$p_{\mathcal{S}}(X, Y) = \begin{cases} 0.5 & Y = 1, -1 \le X < 0, \\ 0.5 & Y = -1, 0 \le X < 1 \end{cases}, \quad p_{\mathcal{T}}(X, Y) = \begin{cases} 0.2 & Y = 1, -1 \le X < 0, \\ 0.2 & Y = -1, 0 \le X < 1 \\ 0.3 & Y = 1, -1 \le X < 1 \\ 0.3 & Y = -1, -1 \le X < 1 \end{cases}, \tag{B.1}$$

*and the hypothesis class $\mathcal{H}$ to be the same as Example 8. Then $\mathcal{S}$ is $(0.5\delta, 0.2\delta)$-transferable (Definition 1) for small $\delta$. However, for any $\Gamma$ that includes the optimal (source and target) classifier $h_0$ we have*

$$\mathrm{T}_\Gamma^{\mathrm{r}}(\mathcal{S}, \mathcal{T}) = \sup_{h \in \Gamma} |\epsilon_{\mathcal{S}}(h) - \epsilon_{\mathcal{T}}(h)| \ge |\epsilon_{\mathcal{S}}(h_0) - \epsilon_{\mathcal{T}}(h_0)| = 0.3. \tag{B.2}$$

The example above shows that when the optimal errors of two domains are dissimilar, simply measuring the difference of errors cannot fully describe the transferability. Instead, we should consider the difference of the *excess risks* as in Definition 1.

## B.2 Other IPMs

Different choices the the function class in (6) could lead to various definitions [46]:

- maximum mean discrepancy (MMD): $\mathcal{F}_{\text{MMD}} = \{f : \|f\|_{\text{Hilbert}} \leq 1\}$ where the norm $\|f\|_{\text{Hilbert}}$ is defined on a reproducing kernel Hilbert space (RKHS).

- Wasserstein distance: $\mathcal{F}_{\text{Wasserstein}} = \{f : \|f\|_L \leq 1\}$ where $\|f\|_L = 1$ is the Lipschitz semi-norm of a real valued function $f$. It is also known as the Kantorovich metric.

- total variation metric: $\mathcal{F}_{\text{TV}} = \{\|f\|_\infty \leq 1\}$ where $\|f\|_\infty = \sup_x\{|f(x)|\}$ is the bound of $f$. This measures the total difference of the probability density functions (PDFs).

- Dudley metric: $\mathcal{F}_{\text{Dudley}} = \{\|f\|_\infty + \|f\|_L \leq 1\}$.

- Kolmogorov distance: $\mathcal{F}_{\text{Kolmogorov}} = \{x \mapsto \mathbb{1}(x \leq t), t \in \mathbb{R}^d\}$ where we have $x \in \mathbb{R}^d$ and $x \leq t$ means that for all components we have $x_i \leq t_i$. This measures the total difference of the cumulative density functions (CDFs).

## B.3 Estimation of transfer measures with VC dimension and Natarajan dimension

In this section, we review Rademacher complexity and show that it can be upper bounded by VC dimension [e.g. 44]. We use $\text{VCdim}(\cdot)$ to represent the VC dimension of a function class. We also show that the estimation error in Lemma 9' can be upper bounded with Natarajan dimension.

**Definition 14** (**Rademacher complexity**). *The Rademacher complexity of an i.i.d. drawn sample set $S = \{w_i\}_{i=1}^m$, over $\mathcal{F}$ is defined as:*

$$\mathfrak{R}_m(\mathcal{F}) = \mathbb{E}_S \left[ \mathbb{E}_{\sigma_i} \sup_{f \in \mathcal{F}} \frac{1}{m} \sum_{i=1}^m \sigma_i f(w_i) \right], \text{ where}$$

$\{\sigma_i\}_{i=1}^m$ *are independently drawn such that* $Pr(\sigma_i = 1) = Pr(\sigma_i = -1) = \dfrac{1}{2}$.

**Lemma 15.** *Denote $d = VCdim(\Gamma)$ where $\Gamma$ is a set of functions taking values $\{-1, +1\}$. For any $m \in \mathbb{N}_+$, we have:*

$$\mathfrak{R}_m(\Gamma) \leq \sqrt{\frac{2}{m} \log \sum_{i=0}^d \binom{m}{i}}, \tag{B.3}$$

*if $m \geq d$, then*

$$\mathfrak{R}_m(\Gamma) \leq \sqrt{\frac{2d}{m} \log \frac{em}{d}}. \tag{B.4}$$

*Proof.* This lemma follows from Corollary 3.8, Theorem 3.17 and Corollary 3.18 of Mohri et al. [34]. $\qquad\square$

Combining Theorem 10' and Lemma 15, we obtain the following corollary:

**Corollary 16.** *Suppose $\widehat{\mathcal{S}}$ and $\widehat{\mathcal{T}}$ are sample distributions of $\mathcal{S}$ and $\mathcal{T}$, with samples drawn i.i.d. Denote the sample numbers of $\widehat{\mathcal{S}}$ and $\widehat{\mathcal{T}}$ are separately $m$ and $k$. If $\mathcal{H}$ is a set of binary classifiers*

*with labels* $\{-1, 1\}$, *then for any* $\Gamma \subseteq \mathcal{H}$ *with* $d = \mathrm{VCdim}(\Gamma)$, *any of the following holds w.p.* $1 - \delta$:

$$\mathrm{T}_\Gamma(\mathcal{S}\|\mathcal{T}) \leq \mathrm{T}_\Gamma(\widehat{\mathcal{S}}\|\widehat{\mathcal{T}}) + 2\sqrt{\frac{2}{m}\log\sum_{i=0}^{d}\binom{m}{i}} + 2\sqrt{\frac{2}{k}\log\sum_{i=0}^{d}\binom{k}{i}} + 2\sqrt{\frac{\log(4/\delta)}{2m}} + 2\sqrt{\frac{\log(4/\delta)}{2k}},$$
(B.5)

$$\mathrm{T}_\Gamma(\mathcal{S}, \mathcal{T}) \leq \mathrm{T}_\Gamma(\widehat{\mathcal{S}}, \widehat{\mathcal{T}}) + 2\sqrt{\frac{2}{m}\log\sum_{i=0}^{d}\binom{m}{i}} + 2\sqrt{\frac{2}{k}\log\sum_{i=0}^{d}\binom{k}{i}} + 2\sqrt{\frac{\log(4/\delta)}{2m}} + 2\sqrt{\frac{\log(4/\delta)}{2k}},$$
(B.6)

$$\mathrm{T}_\Gamma^{\mathrm{r}}(\mathcal{S}, \mathcal{T}) \leq \mathrm{T}_\Gamma^{\mathrm{r}}(\widehat{\mathcal{S}}, \widehat{\mathcal{T}}) + \sqrt{\frac{2}{m}\log\sum_{i=0}^{d}\binom{m}{i}} + \sqrt{\frac{2}{k}\log\sum_{i=0}^{d}\binom{k}{i}} + \sqrt{\frac{\log(4/\delta)}{2m}} + \sqrt{\frac{\log(4/\delta)}{2k}}.$$
(B.7)

*If* $m \geq d$ *and* $k \geq d$, *then any of the following holds w.p.* $1 - \delta$:

$$\mathrm{T}_\Gamma(\mathcal{S}\|\mathcal{T}) \leq \mathrm{T}_\Gamma(\widehat{\mathcal{S}}\|\widehat{\mathcal{T}}) + 2\sqrt{\frac{2d}{m}\log\frac{em}{d}} + 2\sqrt{\frac{2d}{k}\log\frac{ek}{d}} + 2\sqrt{\frac{\log(4/\delta)}{2m}} + 2\sqrt{\frac{\log(4/\delta)}{2k}},$$
(B.8)

$$\mathrm{T}_\Gamma(\mathcal{S}, \mathcal{T}) \leq \mathrm{T}_\Gamma(\widehat{\mathcal{S}}, \widehat{\mathcal{T}}) + 2\sqrt{\frac{2d}{m}\log\frac{em}{d}} + 2\sqrt{\frac{2d}{k}\log\frac{ek}{d}} + 2\sqrt{\frac{\log(4/\delta)}{2m}} + 2\sqrt{\frac{\log(4/\delta)}{2k}},$$
(B.9)

$$\mathrm{T}_\Gamma^{\mathrm{r}}(\mathcal{S}, \mathcal{T}) \leq \mathrm{T}_\Gamma^{\mathrm{r}}(\widehat{\mathcal{S}}, \widehat{\mathcal{T}}) + \sqrt{\frac{2d}{m}\log\frac{em}{d}} + \sqrt{\frac{2d}{k}\log\frac{ek}{d}} + \sqrt{\frac{\log(4/\delta)}{2m}} + \sqrt{\frac{\log(4/\delta)}{2k}}.$$
(B.10)

Moreover, if the hypothesis class $\mathcal{H}$ is the set of all possible functions that can be constructed through a fixed structure ReLU/LeakyReLU network, with $W$ the number of parameters and $L$ the number of layers, then there exists an absolute constant $C$ such that $d \leq CWL\log W$ [7].

A generalization of VC dimension is called Natarajan dimension [38], which coincides with VC dimension when the classification task is binary. We have the following result [44, Theorem 29.3]:

**Lemma 17.** *Suppose the Natarajan dimension of* $\Gamma$ *is* $d$ *and the number of classes is* $K$ *for multiclass classification. There exists absolute constant* $C$ *such that for any domain* $\mathcal{D}$, *with probability* $1 - \delta$ *the following holds:*

$$\mathrm{est}_\Gamma(\mathcal{D}) = \sup_{h\in\Gamma}|\epsilon_\mathcal{D}(h) - \epsilon_{\widehat{\mathcal{D}}}(h)| \leq C\sqrt{\frac{d\log K + \log(1/\delta)}{m}}.$$
(B.11)

With this lemma we have the corollary:

**Corollary 18.** *Suppose the Natarajan dimension of* $\Gamma$ *is* $d$ *and the number of classes is* $K$ *for multiclass classification. Suppose* $\widehat{\mathcal{S}}$ *and* $\widehat{\mathcal{T}}$ *are i.i.d. sample distributions drawn from distributions of* $\mathcal{S}$ *and* $\mathcal{T}$, *with sample number* $m$ *and* $k$, *then w.p. at least* $1 - \delta$ *we have:*

$$\mathrm{T}_\Gamma(\mathcal{S}\|\mathcal{T}) \leq \mathrm{T}_\Gamma(\widehat{\mathcal{S}}\|\widehat{\mathcal{T}}) + 2C\sqrt{\frac{d\log K + \log(2/\delta)}{m}} + 2C\sqrt{\frac{d\log K + \log(2/\delta)}{k}},$$
(B.12)

$$\mathrm{T}_\Gamma(\mathcal{S}, \mathcal{T}) \leq \mathrm{T}_\Gamma(\widehat{\mathcal{S}}, \widehat{\mathcal{T}}) + 2C\sqrt{\frac{d\log K + \log(2/\delta)}{m}} + 2C\sqrt{\frac{d\log K + \log(2/\delta)}{k}},$$
(B.13)

$$\mathrm{T}_\Gamma^{\mathrm{r}}(\mathcal{S}, \mathcal{T}) \leq \mathrm{T}_\Gamma^{\mathrm{r}}(\widehat{\mathcal{S}}, \widehat{\mathcal{T}}) + C\sqrt{\frac{d\log K + \log(2/\delta)}{m}} + C\sqrt{\frac{d\log K + \log(2/\delta)}{k}}.$$
(B.14)

*Proof.* This proof is similar to the proof of Theorem 10', using union bound as in (A.33). □

Estimation of Natarajan dimension can be found in Natarajan [38], Shalev-Shwartz and Ben-David [44].

## B.4 Functional point of view of surrogate loss

In this appendix we study the Lipschitzness and strong convexity of the surrogate loss, especially cross entropy. We use the terms distribution and measure interchangeably, since distributions can be treated as probability measures.

### B.4.1 Lipschitz continuity of loss

Let define the $L_p$ distance ($p \geq 1$) (e.g. Rudin [42]) between two functions:

$$\|h - h'\|_{p,\mu} = \left( \int \|h(x) - h'(x)\|_2^p d\mu \right)^{1/p}, \tag{B.15}$$

where $\mu$ is a measure. We consider the following definition of Lipschitz functional:

**Definition 19** (**Lipschitz continuity**). *A functional $h \mapsto f(h)$ that maps a function to a real number is $f$ is Lipschitz continuous on $\mathcal{H}$ w.r.t. measure $\mu$ if there exists an absolute constant $L$ such that:*

$$|f(h) - f(h')| \leq L\|h - h'\|_{1,\mu} \tag{B.16}$$

*for all function $h, h' \in \mathcal{H}$.*

One can show that the cross entropy loss is a Lipschitz continuous functional with mild assumptions:

**Proposition 20.** *For binary classification with labels $\{-1, +1\}$, suppose $\mathcal{H}$ is a hypothesis class whose elements satisfy $h : \mathcal{X} \to (-1 + \delta, 1 - \delta)$ with $0 < \delta < 1$, then $\epsilon_{\mathcal{D}}^{\mathrm{CE}}$ is $(\log 2)^{-1}\delta^{-1}$ Lipschitz continuous w.r.t. any distribution $\mathcal{D}$. Furthermore, for multi-class classification, suppose $\mathcal{Y} = \{1, 2, \ldots, K\}$ and the prediction $h(x)$ is a $K$-dimensional probability vector on the simplex. If $\mathcal{H}$ is a hypothesis class whose elements satisfy $h_i(x) \geq \delta$ for all $i \in \mathcal{Y}$ and $x \in \mathcal{X}$, then $\epsilon_{\mathcal{D}}^{\mathrm{CE}}$ is $(\log 2)^{-1}\delta^{-1}$ Lipschitz continuous w.r.t. any distribution $\mathcal{D}$.*

Note that a simplex is defined as: $\{\pi \in \mathbb{R}^d : \mathbb{1}^\top \pi = 1, \pi_i \geq 0\}$, where $\pi$ is called a probability vector. Before we move on to the proof, we can show that the assumption of $h$ is often satisfied in practice. For binary classification, the widely used tanh/sigmoid function can guarantee that the value of $h$ is never exactly $-1$ or $1$. For multiclass classification, the softmax function guarantees that $h_i(x) > 0$ for all $i$ and $x \in \mathcal{X}$. If the input space is bounded and $h$ is continuous, then $h_i(x) \geq \delta$ for all $i$ and $x \in \mathcal{X}$.

*Proof.* For binary classification we have:

$$\epsilon_{\mathcal{D}}^{\mathrm{CE}}(h) = \int p_{\mathcal{D}}(x, 1)\ell^{\mathrm{CE}}(h(x), 1) + p_{\mathcal{D}}(x, -1)\ell^{\mathrm{CE}}(h(x), -1)dx$$

$$= \int -p_{\mathcal{D}}(x, 1)\log_2 \frac{1 + h(x)}{2} - p_{\mathcal{D}}(x, -1)\log_2 \frac{1 - h(x)}{2}dx. \tag{B.17}$$

Therefore, with the mean value theorem we have:

$$|\epsilon_{\mathcal{D}}^{\mathrm{CE}}(h) - \epsilon_{\mathcal{D}}^{\mathrm{CE}}(h')| = (\log 2)^{-1} \left| \int (h(x) - h'(x)) \left( \frac{-p_{\mathcal{D}}(x, 1)}{1 + h_\xi(x)} + \frac{p_{\mathcal{D}}(x, -1)}{1 - h_\xi(x)} \right) dx \right|,$$

$$\leq (\log 2)^{-1} \int |h(x) - h'(x)| \left| \frac{-p_{\mathcal{D}}(x, 1)}{1 + h_\xi(x)} + \frac{p_{\mathcal{D}}(x, -1)}{1 - h_\xi(x)} \right| dx$$

$$\leq (\log 2)^{-1} \int |h(x) - h'(x)| \left( \left| \frac{-p_{\mathcal{D}}(x, 1)}{1 + h_\xi(x)} \right| + \left| \frac{p_{\mathcal{D}}(x, -1)}{1 - h_\xi(x)} \right| \right) dx$$

$$\leq (\log 2)^{-1} \int |h(x) - h'(x)|\delta^{-1}(p_{\mathcal{D}}(x, 1) + p_{\mathcal{D}}(x, -1))dx$$

$$= (\log 2)^{-1}\delta^{-1}\|h - h'\|_{1,\mathcal{D}}. \tag{B.18}$$

where in the first line $h_\xi(x) = (1 - \xi(x))h(x) + \xi(x)h'(x)$ is a (pointwise) convex combination of $h(x)$ and $h'(x)$ with $0 \leq \xi(x) \leq 1$; in the third line we used triangle inequality; in the fourth line we use the condition that the values of $h, h'$ are in the region $(-1 + \delta, 1 - \delta)$.

Similarly, for multiclass classification with $K$ classes, the ground truth $y$ is a one-hot $K$-dimensional vector, and the prediction $h(x)$ is a $K$-dimensional vector on a simplex. The cross entropy loss is:

$$\epsilon_{\mathcal{D}}^{\mathrm{CE}}(h) = \sum_y \int \ell^{\mathrm{CE}}(h(x), y)p(x,y)dx = \sum_y \int -y \cdot \log_2 h(x)p(x,y)dx. \tag{B.19}$$

Similarly, we have:

$$\begin{aligned}
|\epsilon_{\mathcal{D}}^{\mathrm{CE}}(h) - \epsilon_{\mathcal{D}}^{\mathrm{CE}}(h')| &= (\log 2)^{-1} \left| \sum_y \int -y \cdot \frac{h(x) - h'(x)}{h_\xi(x)} p_{\mathcal{D}}(x,y)dx \right| \\
&\le (\log 2)^{-1} \sum_y \int \left| -y \cdot \frac{h(x) - h'(x)}{h_\xi(x)} \right| p_{\mathcal{D}}(x,y)dx \\
&\le (\log 2)^{-1}\delta^{-1} \sum_y \int \|y\|_2 \cdot \|h(x) - h'(x)\|_2 p_{\mathcal{D}}(x,y)dx \\
&= (\log 2)^{-1}\delta^{-1}\|h - h'\|_{1,\mathcal{D}}, \tag{B.20}
\end{aligned}$$

where in the first line we use the mean value theorem and $h_\xi(x) = (1 - \xi(x))h(x) + \xi(x)h'(x)$ is a (pointwise) convex combination of $h(x)$ and $h'(x)$ with $0 \le \xi(x) \le 1$; also in the first line we define $(h(x) - h'(x))/h_\xi(x)$ to be a vector with each component to be $(h_i(x) - h_i'(x))/h_\xi(x)_i$; in the third line we use Cauchy–Schwarz inequality and that $h_i(x) \ge \delta$, $h_i'(x) \ge \delta$ for any $i$ and any $x \in \mathcal{X}$. $\quad\square$

### B.4.2 Strongly convex functional

So far, we have seen that for Lipschitz continuous loss, if the change of $h$ is small, then the change of loss $\epsilon_{\mathcal{D}}(h)$ is also small. Now we ask if the converse is true. This is important to characterize the $\delta$-minimal set (the set of approximately optimal classifiers). We first define strongly convex functional:

**Definition 21.** *A functional $f : \mathcal{H} \to \mathbb{R}$ is $\lambda$-strongly convex on a convex set $\mathcal{H}$ w.r.t. measure $\mu$ if for any $h, h' \in \mathcal{H}$ and $\alpha \in [0,1]$, we have:*

$$f(\alpha h + (1-\alpha)h') \le \alpha f(h) + (1-\alpha)f(h') - \frac{\lambda}{2}\alpha(1-\alpha)\|h - h'\|_{2,\mu}^2, \tag{B.21}$$

*where we defined the $L_2$ norm of a function:*

$$\|h - h'\|_{2,\mu} := \left( \int \|h - h'\|_2^2 d\mu \right)^{1/2}. \tag{B.22}$$

We use $L_2$ norm because it can translate the strong convexity of the loss functional to the strong convexity of the loss function $\ell(\cdot, y)$ easily:

**Lemma 22.** *Given a convex hypothesis class $\mathcal{H}$, suppose that $\ell$ is $\lambda$-strongly convex in the first argument, i.e. for any $y \in \mathcal{Y}$, $\hat{y}_1, \hat{y}_2$ and $\alpha \in [0,1]$ we have:*

$$\ell(\alpha \hat{y}_1 + (1-\alpha)\hat{y}_2, y) \le \alpha \ell(\hat{y}_1, y) + (1-\alpha)\ell(\hat{y}_2, y) - \frac{\lambda}{2}\alpha(1-\alpha)\|\hat{y}_1 - \hat{y}_2\|_2^2, \tag{B.23}$$

*then the loss functional*

$$\epsilon_{\mathcal{D}}(h) = \sum_y \int p_{\mathcal{D}}(x,y)\ell(h(x), y)dx \tag{B.24}$$

*is also $\lambda$-strongly convex w.r.t. measure $\mathcal{D}$.*

*Proof.* Straightforward by plugging in Definition 21. $\quad\square$

For cross entropy loss, we have the following:

**Corollary 23.** *For binary classification, cross entropy risk functional $\epsilon_{\mathcal{D}}^{\mathrm{CE}}$ is $(4\log 2)^{-1}$-strongly convex on $\mathcal{D}$ and $(\log 2)^{-1}$-strongly convex on $\mathcal{D}$ for multiclass classification.*

*Proof.* For binary classification, we have:

$$\ell^{\mathrm{CE}}(\hat{y}, 1) = -\log_2 \frac{1+\hat{y}}{2}, \ \ell^{\mathrm{CE}}(\hat{y}, -1) = -\log_2 \frac{1-\hat{y}}{2}, \tag{B.25}$$

which are both $(4\log 2)^{-1}$-strongly convex on $\hat{y} \in (-1, 1)$. For multiclass classification, we have:

$$\ell^{\mathrm{CE}}(\hat{y}, y) = -\log_2 \hat{y}_i, \tag{B.26}$$

for any unit one-hot vector $y = e_i$ ($e_i$ is the $i^{\mathrm{th}}$ element of standard basis in $\mathbb{R}^K$). This is $(\log 2)^{-1}$-strongly convex for $\hat{y}_i \in (0, 1)$. The rest follows from Lemma 22. $\qquad\square$

From the strongly convexity we can derive the uniqueness of the function (up to $L_2$ norm) and relate $\delta$-minimal set to an $L_2$ neighborhood of an optimal classifier.

**Theorem 24.** *For any $\lambda$-strongly convex functional $f$ on a convex hypothesis class $\mathcal{H}$ w.r.t. measure $\mu$, the minimizer is almost surely unique, in the sense that if $h_1^*$, $h_2^*$ are both minimizers, then*

$$\|h_1^* - h_2^*\|_{2,\mu} = 0, \tag{B.27}$$

*and thus $h_1^*$, $h_2^*$ only differ by a measure zero set. Suppose $h^* \in \arg\min f(h)$. If $f(h) \leq f^* + \epsilon$ with $f^*$ the optimal value, then*

$$\|h - h^*\|_{2,\mu} \leq \sqrt{\frac{2}{\lambda}\epsilon}. \tag{B.28}$$

*Proof.* It suffices to prove the second claim only. From the definition of strong convexity, for $\alpha \in [0, 1]$ we have:

$$f^* \leq f(\alpha h^* + (1-\alpha)h) \leq \alpha f(h^*) + (1-\alpha)f(h) - \frac{\lambda}{2}\alpha(1-\alpha)\|h^* - h\|_{2,\mu}^2$$

$$\leq (1-\alpha)\epsilon + f^* - \frac{\lambda}{2}\alpha(1-\alpha)\|h^* - h\|_{2,\mu}^2, \tag{B.29}$$

where we use $h^* \in \arg\min f(h)$ and $f(h) \leq f^* + \epsilon$. From this inequality we obtain that:

$$\|h - h^*\|_{2,\mu}^2 \leq \frac{2\epsilon}{\alpha\lambda}. \tag{B.30}$$

By taking $\alpha \to 1$ we obtain (B.28). $\qquad\square$

With this theorem we can characterize the $\delta$-minimal set $\arg\min(\epsilon_{\mathcal{D}}, \delta)$ as some neighborhood of the unique optimal classifier $h^*$, if the functional $\epsilon_{\mathcal{D}}$ is strongly convex and Lipschitz continuous. Symbolically, it can be represented as:

$$\mathcal{B}_2(h^*) \subseteq \arg\min(\epsilon_{\mathcal{D}}, \delta) \subseteq \mathcal{B}_1(h^*), \tag{B.31}$$

where $\mathcal{B}_p(h^*)$ is some $L_p$ norm ball with the center $h^*$.

### B.4.3 Parametric formulation of classifier

We discussed the $L_p$ distance between functions in previous subsections. In practice the functions are often parametrized:

$$h(x) = q(\theta, x). \tag{B.32}$$

One can show that $L_p$ distances between two functions $h = q(\theta, \cdot)$ and $h' = q(\theta', \cdot)$ on the function space can be upper bounded:

**Theorem 25.** *Suppose $h = q(\theta, \cdot)$ is parameterized by $\theta$ and for any $x \in \mathcal{X}$, $q(\cdot, x)$ is $L$-Lipschitz continuous (w.r.t. $\ell_2$ norm), then for any $1 \leq p < \infty$ and probability measure $\mu$ we have:*

$$\|h - h'\|_{p,\mu} \leq L\|\theta - \theta'\|_2. \tag{B.33}$$

*Proof.* From the Lipschitz continuity we can derive:

$$\|h - h'\|_{p,\mu} = \left( \int \|h(x) - h'(x)\|_2^p d\mu \right)^{1/p}$$

$$= \left( \int \|q(\theta, x) - q(\theta', x)\|_2^p d\mu \right)^{1/p}$$

$$\leq \left( \int (L\|\theta - \theta'\|_2)^p d\mu \right)^{1/p}$$

$$= L\|\theta - \theta'\|_2. \tag{B.34}$$

$\square$

The theorem above tells us that in parametrized models the closeness in terms of parameters can imply the closeness in terms of the model function. However, the converse may not be true. For example, we can permute hidden neurons of the same layer in a neural network and obtain the same function, but the parametrization can be drastically different.

## B.5 Comparison with other frameworks

We compare our Algorithm 2 with existing adversarial training frameworks.

**Distributional robustness optimization (DRO)** Sinha et al. [45] proposed a distributional robustness framework for generalizing to unseen domains. In this framework, the following minimax problem is proposed:

$$\min_{g,h} \max_{\mathcal{S}':W(\mathcal{S}',\mathcal{S})\leq\delta} \epsilon_{\mathcal{S}}'(h \circ g), \tag{B.35}$$

which says that the classification error is small for any distribution $\mathcal{S}'$ close to our original source distribution $\mathcal{S}$. Here $W(\cdot,\cdot)$ denotes the Wasserstein metric. As we have discussed in Example 8, transferability does not necessarily mean that the distributions have to be close.

**DANN** The Domain Adversarial Neural Network (DANN) formulation [16] solves the following minimax optimization problem:

$$\min_{g,h} \max_{h'} \epsilon_{\mathcal{S}}(h \circ g) + \mathbb{E}_{x \sim p_{\mathcal{S}}|_x}[\log(h' \circ g)(x)] + \mathbb{E}_{x \sim p_{\mathcal{T}}|_x}[\log(1 - (h' \circ g)(x))], \tag{B.36}$$

where $g$ is a feature embedding, $h$ is a classifier and $h'$ is a domain discriminator. If we can solve the inner maximization problem exactly, then we obtain the Jensen–Shannon divergence between the push-forwards of the input distributions $g\#p_S|_x$ and $g\#p_T|_x$. In other words, we want to obtain a feature embedding $g$ and a classifier $h$ such that:

$$\epsilon_{\mathcal{S}}(h \circ g) + D_{\mathrm{JS}}((g\#p_{\mathcal{S}}|_x)\|(g\#p_{\mathcal{T}}|_x)), \tag{B.37}$$

is minimized, with $D_{\mathrm{JS}}$ denoting the Jensen–Shannon divergence. On the one hand, we need to have small classification error given the feature embedding $g$. On the other hand, the feature embedding between source and target should be similar. Our framework is similar to DANN in the sense that they both solve minimax problems. The difference is that we minimize the transfer measure which is weaker than the similarity between distributions (Example 8).

$\mathcal{H}\Delta\mathcal{H}$**-divergence** Finally we prove that our transfer measure is tighter than $\mathcal{H}\Delta\mathcal{H}$-divergence [12]. We rewrite the theoretical result regarding $\mathcal{H}\Delta\mathcal{H}$-divergence:

**Theorem 26** (Theorem 1, [12]). *Let* $\lambda^* = \mathrm{argmin}_{h\in\mathcal{H}}(\epsilon_{\mathcal{T}}(h) + \epsilon_{\mathcal{S}}(h))$, *and the* $\mathcal{H}\Delta\mathcal{H}$*-divergence between the input marginal distributions* $\mathcal{S}|_x$ *and* $\mathcal{T}|_x$ *to be* $d_{\mathcal{H}\Delta\mathcal{H}}(\mathcal{S}|_x, \mathcal{T}|_x)$, *then for binary classification and for any* $h \in \mathcal{H}$ *we have:*

$$\epsilon_{\mathcal{T}}(h) \leq \epsilon_{\mathcal{S}}(h) + \lambda^* + \frac{1}{2}d_{\mathcal{H}\Delta\mathcal{H}}(\mathcal{S}|_x, \mathcal{T}|_x). \tag{B.38}$$

Now let us prove that our Proposition 3 is tighter than Theorem 26:

**Proposition 27.** *The target error bound with our transfer measure* $\mathrm{T}_\Gamma(\mathcal{S}\|\mathcal{T})$ *is tighter than the target error bound with* $\mathcal{H}\Delta\mathcal{H}$*-divergence, i.e., for any* $h \in \Gamma$ *we have:*

$$\epsilon_\mathcal{T}(h) \leq \epsilon_\mathcal{S}(h) + \epsilon_\mathcal{T}^* - \epsilon_\mathcal{S}^* + \mathrm{T}_\Gamma(\mathcal{S}\|\mathcal{T}) \leq \epsilon_\mathcal{S}(h) + \lambda^* + \frac{1}{2}d_{\mathcal{H}\Delta\mathcal{H}}(\mathcal{S}|_x, \mathcal{T}|_x). \tag{B.39}$$

*Proof.* Note that from Definition 2 we can rewrite the middle of (B.39) as $\sup_{h\in\Gamma}(\epsilon_\mathcal{T}(h) - \epsilon_\mathcal{S}(h))$. Suppose $h^* \in \arg\max_{h\in\Gamma}(\epsilon_\mathcal{T}(h) - \epsilon_\mathcal{S}(h))$, then from Theorem 26 we have:

$$\epsilon_\mathcal{T}(h^*) \leq \epsilon_\mathcal{S}(h^*) + \lambda^* + \frac{1}{2}d_{\mathcal{H}\Delta\mathcal{H}}(\mathcal{S}|_x, \mathcal{T}|_x), \tag{B.40}$$

and thus:

$$\epsilon_\mathcal{T}^* - \epsilon_\mathcal{S}^* + \mathrm{T}_\Gamma(\mathcal{S}\|\mathcal{T}) = \sup_{h\in\Gamma}(\epsilon_\mathcal{T}(h) - \epsilon_\mathcal{S}(h)) = \epsilon_\mathcal{T}(h^*) - \epsilon_\mathcal{S}(h^*) \leq \lambda^* + \frac{1}{2}d_{\mathcal{H}\Delta\mathcal{H}}(\mathcal{S}|_x, \mathcal{T}|_x). \tag{B.41}$$

$\square$

## C   Additional Experiments

We present additional experimental details in this section.

### C.1   Datasets

The four datasets in this paper are RotatedMNIST [18], PACS [28], Office-Home [51] and WILDS-FMoW [23]. Here is a short description:

- RotatedMNIST: this dataset is an adaptation of MNIST. It has six domains, and each domain rotates the images in MNIST with a different angle. The angles are $\{0°, 15°, 30°, 45°, 60°, 75°\}$. We choose the domain with $0°$ to be the target domain and the rest to be the source domains. Each image is grayscale and has $28 \times 28$ pixels. The label set is $\{0, 1, \ldots, 9\}$. The numbers of images of each domain are 11667, 11667, 11667, 11667, 11666, 11666. The total is 70000.

- PACS: this dataset has four domains: photo (P), art painting (A), cartoon (C) and sketch (S). Each image is RGB colored and has $224 \times 224$ pixels. There are 7 categories in total and 9991 images. The number of images of each domain: A: 2048; C: 2344; P: 1670; S: 3929. We choose the art painting domain to be the target domain and the rest to be the source domains.

- Office-Home: this dataset has four domains: Art, Clipart, Product, Real-World. Each image is RGB colored and has $224 \times 224$ pixels. There are 65 categories and 15588 images in total. The numbers of images of each domain: Art: 2427, Clipart: 4365, Product: 4439, Real-World: 4357. We choose the Art domain to be the target domain and the rest to be the source domains.

- WILDS-FMoW: WILDS [23] is a benchmark for domain generalization including several datasets. The Functional Map of the World (FMoW) is one of them, which is a variant of Christie et al. [14]. Each image is RGB colored and has $224 \times 224$ pixels. There are 62 categories and 469835 images in total. There six domains in total and we choose five of them, since the last domain has too few images. The numbers of images of each domain are 103299, 162333, 33239, 157711, 13253, and we choose the last domain as the target domain. The rest are source domains. The license can be found at `https://wilds.stanford.edu/datasets/`.

### C.2   Experimental settings

We introduce the experimental settings in this subsection. The code is modified from `https://github.com/facebookresearch/DomainBed`, with the license in `https://github.com/facebookresearch/DomainBed/blob/master/LICENSE`.

- Hardware: Our experiments are run on a cluster of GPUs, including NVIDIA RTX6000, T4 and P100.

- Datasplit: we use the same data split as in Gulrajani and Lopez-Paz [20] except the WILDS-FMoW dataset, where we throw away the last region because it has only very few samples (201 samples). For all datasets we use data augmentation.

- Batch size: for all experiments on RotatedMNIST we choose batch size 64, for Office-Home and PACS we choose batch size 32 (for our Transfer algorithm and PACS we choose batch size 16), and for WILDS-FMoW we choose batch size 16. In each epoch, we go through $k$ steps, where $k$ is the smallest number of samples among domains, divided by the batch size.

- Optimization: for the training of all other algorithms different from our Transfer Algorithm, we use the default setting from Gulrajani and Lopez-Paz [20]. We choose Adam as the default optimizer for training, with learning rate 1e-3 for RotatedMNIST, and learning rate 5e-5 for other datasets. For RotatedMNIST, PACS and Office-Home we run for $5000$ steps; For WILDS-FMoW we run for $50000$ steps.

- Neural Architecture: we use the same neural architecture as in Gulrajani and Lopez-Paz [20]. For each dataset, the feature embedding and classifier architectures for all algorithms are the same. Specifically, all classifiers are linear layers. For RotatedMNIST the feature embedding is CNN with batch normalization and for other datasets the feature embedding is ResNet50.

- Algorithm 1: we choose Adam optimizer with projection. The learning rates are the same as the training algorithms: for RotatedMNIST we choose 1e-3, and we choose 5e-5 for others. We run the algorithm for 10 epochs and for three independent trials. Among the three trials, we choose the accuracies with the largest gap between the target domain and one of the source domains. The source domain is chosen in such a way that the gap is the largest among all source domains.

- Algorithm 2 optimization: for RotatedMNIST we run Adam for minimization with learning rate 0.01 and Stochastic Gradient Ascent (SGA) for maximization with learning rate 0.01. We choose the ascent steps to be 30 for each inner loop and the projection radius to be $\delta = 10.0$; for PACS we run Adam for minimization with learning rate 5e-5 and Stochastic Gradient Ascent (SGA) for maximization with learning rate 0.001. We choose the ascent steps to be 30 for each inner loop and the projection radius to be $\delta = 0.3$; for Office-Home dataset we load the pretrained model from SD, and run Stochastic Gradient Descent Ascent with learning rate 0.001 and $\delta = 0.3$, i.e., each inner loop takes only one step of SGA and each outer loop takes one step of SGD; for WILDS-FMoW dataset we loaded the pretrained model from ERM, and run SGA for 20 steps in each inner loop, with learning rate 0.001 and $\delta = 0.5$, for each outer loop we run SGD with `lr` $= 0.001$.

- Step number for Algorithm 2: for RotatedMNIST and PACS we train for $8000$ outer steps with each outer step including 30 inner steps. For Office-Home we train for $5000$ outer steps with each outer step including one inner step; for WILDS-FMoW we train for $5000$ outer loops with each outer step including 20 inner steps.

## C.3   Additional results

We present additional experiments on RotatedMNIST [18], PACS [28] and Office-Home [51]. Thanks to the suite from Gulrajani and Lopez-Paz [20], we are able to compare a wide array of algorithms under the same settings. The algorithms that we compare include

- Empirical Risk Minimization [ERM, 50]
- Invariant Risk Minimization [IRM, 3]
- Domain Adversarial Neural Network [DANN, 16]
- Conditional DANN [CDANN, 31]
- Correlation Alignment [CORAL, 47]
- Maximum Mean Discrepancy [MMD, 30]
- Variance Risk Extrapolation [VREx, 26]

- Mariginal Transfer Learning [MTL, 11]

- Spectral Decoupling [SD, 40]

- Meta Learning Domain Generalization [MLDG, 29]

- Mixup [53, 54]

- Representation Self-Challenging [RSC, 22]

- Group Distributionally Robust Optimization [GroupDRO, 43]

- Style-Agnostic Network [SagNet, 37]

### C.3.1  RotatedMNIST

In Figure 4 we show the performance of various algorithms on RotatedMNIST, including ERM, IRM, DANN, CORAL, MMD, VREx, MTL, SD and our Transfer algorithm. It can be seen that many algorithms fail our attack. For instance, based on the learned features, MTL classifies a source domain with $\sim$95% (at $\delta = 3.0$) but the target accuracy drops by $\sim$20%.

We also compare our Transfer algorithm (Algorithm 2) with different hyperparameters. From Figure 5 we can see that for RotatedMNIST, taking more inner steps (per outer step) has better performance.

Finally, we present results from Algorithm 1 with information about losses and accuracies, for a wide array of algorithms in Table 2.

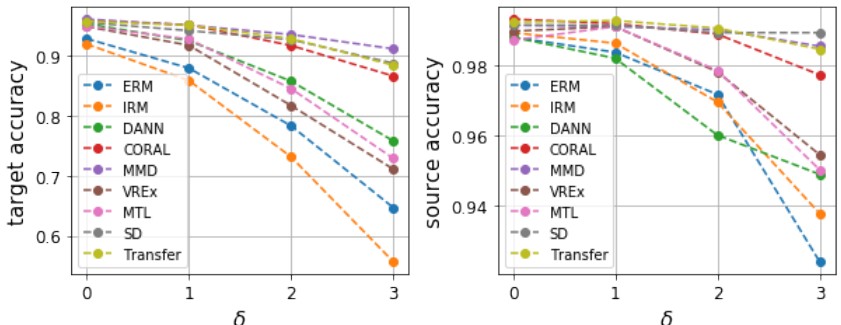

Figure 4: Measuring the transferability of various algorithms for domain generalization on RotatedMNIST. For the Transfer algorithm we take $\delta = 10.0$ and the number of ascent steps to be 30.

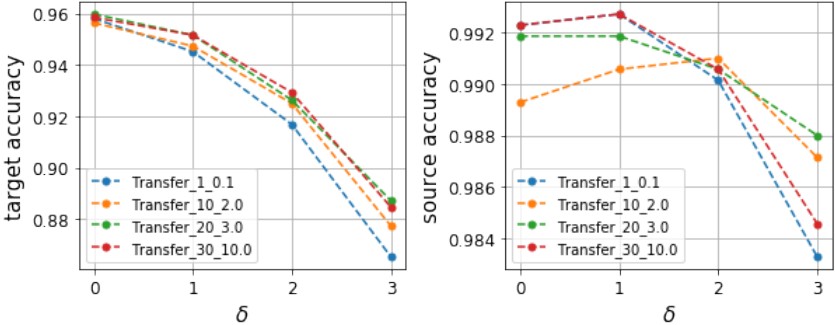

Figure 5: Evaluation of transferability of popular algorithms for domain generalization on Transfer algorithm with different hyperparameters. The dataset is RotatedMNIST. For the ascent method we use SGD with learning rate $0.01$, and the descent method to be Adam with learning rate $10^{-3}$. "`Transfer_d_δ`" means the inner loop takes $d$ steps with the radius $\delta$.

Table 2: Evaluation of transferability of popular algorithms for domain generalization on RotatedM-NIST. **algorithm**: the model that we evaluate; $\delta$: the adversarial radius $\delta$ we choose in Algorithm 1; **max/min index**: the index of the domain with the maximal/minimal (test) classification errors (w.r.t. 0-1 loss), and index $0$ denotes the target domain; **max/min loss**: the largest/smallest loss among domains (including the target domain); **worst/best acc**: the smallest/largest classification test accuracies among domains (including the target domain). All the algorithms are using the same architectures for the feature embedding and the classifier.

| algorithm | $\delta$ | max index | min index | max loss | min loss | worst acc | best acc |
|-----------|------|-----------|-----------|----------|----------|-----------|----------|
| ERM | 0.0 | 0 | 4 | 0.229 | 0.003 | 92.93% | 98.80% |
| ERM | 2.0 | 0 | 4 | 0.975 | 0.083 | 78.61% | 97.17% |
| GroupDRO | 0.0 | 0 | 4 | 0.136 | 0.000 | 95.76% | 99.27% |
| GroupDRO | 2.0 | 0 | 4 | 0.370 | 0.015 | 84.48% | 98.07% |
| SagNet | 0.0 | 0 | 4 | 0.109 | 0.000 | 96.61% | 99.36% |
| SagNet | 2.0 | 0 | 4 | 0.222 | 0.008 | 91.30% | 98.67% |
| IRM | 0.0 | 0 | 4 | 0.578 | 0.263 | 81.87% | 92.20% |
| IRM | 2.0 | 0 | 4 | 1.759 | 0.637 | 46.29% | 86.76% |
| DANN | 0.0 | 0 | 5 | 0.136 | 0.014 | 95.41% | 98.29% |
| DANN | 2.0 | 0 | 5 | 0.441 | 0.098 | 85.81% | 96.19% |
| ARM | 0.0 | 0 | 4 | 0.145 | 0.002 | 95.76% | 99.10% |
| ARM | 2.0 | 0 | 4 | 0.523 | 0.047 | 84.23% | 98.54% |
| Mixup | 0.0 | 0 | 4 | 0.175 | 0.009 | 94.98% | 99.36% |
| Mixup | 2.0 | 0 | 4 | 0.701 | 0.035 | 73.98% | 98.71% |
| CORAL | 0.0 | 0 | 4 | 0.119 | 0.001 | 95.93% | 99.31% |
| CORAL | 2.0 | 0 | 4 | 0.230 | 0.005 | 91.77% | 98.89% |
| CORAL | 3.0 | 0 | 4 | 0.372 | 0.056 | 86.67% | 97.73% |
| MMD | 0.0 | 0 | 3 | 0.125 | 0.005 | 96.19% | 99.14% |
| MMD | 2.0 | 0 | 3 | 0.199 | 0.014 | 93.61% | 99.01% |
| MMD | 3.5 | 0 | 3 | 0.300 | 0.036 | 89.54% | 97.86% |
| RSC | 0.0 | 0 | 4 | 0.146 | 0.000 | 95.46% | 99.31% |
| RSC | 1.0 | 0 | 4 | 0.360 | 0.007 | 89.33% | 98.71% |
| RSC | 2.0 | 0 | 4 | 1.343 | 0.289 | 72.01% | 92.11% |
| VREx | 0.0 | 0 | 5 | 0.137 | 0.003 | 94.94% | 98.97% |
| VREx | 2.0 | 0 | 5 | 0.551 | 0.082 | 81.74% | 97.81% |
| CDANN | 0.0 | 0 | 5 | 0.121 | 0.010 | 95.97% | 98.76% |
| CDANN | 2.0 | 0 | 5 | 0.410 | 0.079 | 84.78% | 95.67% |
| MLDG | 0.0 | 0 | 5 | 0.151 | 0.000 | 95.63% | 98.89% |
| MLDG | 2.0 | 0 | 5 | 0.351 | 0.006 | 88.90% | 98.76% |
| MTL | 0.0 | 0 | 4 | 0.150 | 0.000 | 94.98% | 99.44% |
| MTL | 2.0 | 0 | 4 | 0.417 | 0.014 | 84.57% | 98.20% |
| SD | 0.0 | 0 | 2 | 0.250 | 0.092 | 95.63% | 99.01% |
| SD | 2.0 | 0 | 2 | 0.630 | 0.490 | 92.76% | 98.97% |
| SD | 3.0 | 0 | 2 | 1.070 | 0.937 | 88.81% | 98.33% |

### C.3.2 PACS

We implement similar experiments on PACS. Figure 6 and Table 3 show the results of Algorithm 1. Figure 7 shows that taking more inner steps has better performance.

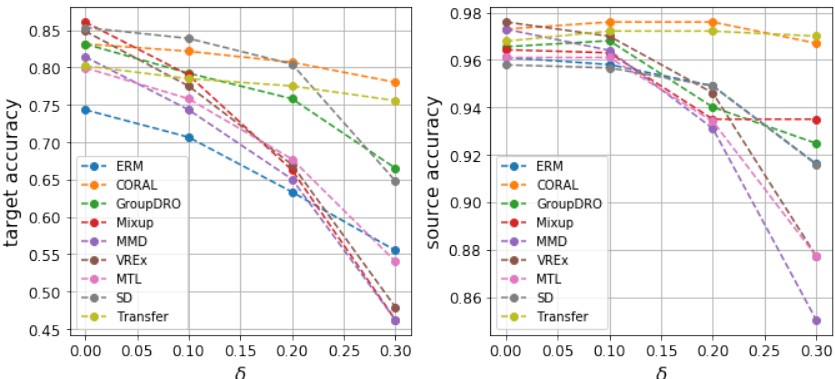

Figure 6: Measuring the transferability of various algorithms for domain generalization on PACS dataset. For the Transfer algorithm we choose $\delta = 0.3$, batch size 16, the number of ascent steps to be 30 using SGD with learning rate 0.001.

### C.3.3 Office-Home

We present results from Algorithm 1 with information about losses and accuracies, for a wide array of algorithms in Table 4 for Office-Home. It can be seen that CORAL and SD learn more robust classifiers while other algorithms are not quite transferable: with a small decrease of source accuracy the target accuracy drops significantly.

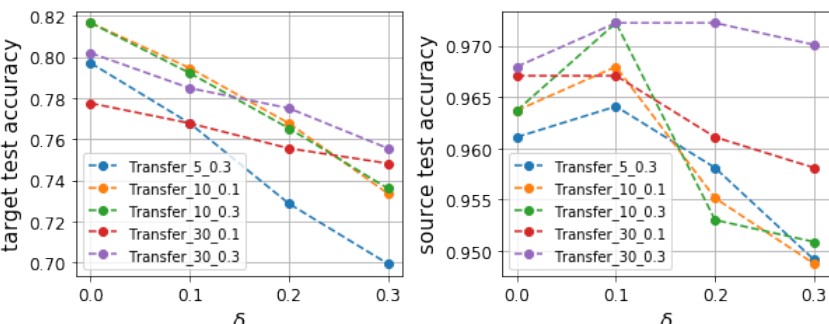

Figure 7: Evaluation of transferability of popular algorithms for domain generalization on Transfer algorithm with different hyperparameters. The dataset is PACS. One can see that if the number of inner steps is large and $\delta$ is large, then the classifier is more robust. "Transfer_$d$_$\delta$" means the inner loop takes $d$ steps with the radius $\delta$.

Table 3: Evaluation of transferability of popular algorithms for domain generalization on PACS. **algorithm**: the model that we evaluate; $\delta$: the adversarial radius $\delta$ we choose in Algorithm 1; **max/min index**: the index of the domain with the maximal/minimal (test) classification errors (w.r.t. 0-1 loss); **max/min loss**: the largest/smallest loss among domains (including the target domain); **worst/best acc**: the smallest/largest classification test accuracies among domains (including the target domain).

| algorithm | $\delta$ | max index | min index | max loss | min loss | worst acc | best acc |
|---|---|---|---|---|---|---|---|
| ERM | 0.0 | 0 | 2 | 1.327 | 0.011 | 74.33% | 96.11% |
| ERM | 0.2 | 0 | 2 | 2.449 | 0.064 | 63.33% | 94.91% |
| GroupDRO | 0.0 | 0 | 2 | 0.820 | 0.012 | 83.13% | 97.60% |
| GroupDRO | 0.2 | 0 | 2 | 1.509 | 0.052 | 75.79% | 95.81% |
| SagNet | 0.0 | 0 | 2 | 0.919 | 0.002 | 77.51% | 99.10% |
| SagNet | 0.1 | 0 | 2 | 1.409 | 0.014 | 71.39% | 97.01% |
| SagNet | 0.2 | 0 | 2 | 2.002 | 0.094 | 60.64% | 94.31% |
| Mixup | 0.0 | 0 | 2 | 0.471 | 0.009 | 86.06% | 99.70% |
| Mixup | 0.1 | 0 | 2 | 0.681 | 0.016 | 78.97% | 98.80% |
| Mixup | 0.2 | 0 | 2 | 0.974 | 0.067 | 66.26% | 96.41% |
| CORAL | 0.0 | 0 | 2 | 0.743 | 0.006 | 83.13% | 97.31% |
| CORAL | 0.2 | 0 | 2 | 0.954 | 0.008 | 80.68% | 97.60% |
| CORAL | 0.3 | 0 | 2 | 1.147 | 0.012 | 78.00% | 96.71% |
| MMD | 0.0 | 0 | 2 | 0.776 | 0.005 | 81.42% | 97.31% |
| MMD | 0.1 | 0 | 2 | 1.203 | 0.006 | 74.33% | 96.41% |
| MMD | 0.2 | 0 | 2 | 1.832 | 0.066 | 65.04% | 93.11% |
| RSC | 0.0 | 0 | 2 | 1.089 | 0.003 | 77.75% | 95.81% |
| RSC | 0.1 | 0 | 2 | 2.535 | 0.129 | 63.81% | 93.41% |
| RSC | 0.2 | 0 | 2 | 4.732 | 0.560 | 43.52% | 82.63% |
| VREx | 0.0 | 0 | 2 | 0.593 | 0.002 | 84.84% | 97.60% |
| VREx | 0.1 | 0 | 2 | 0.912 | 0.009 | 77.51% | 97.01% |
| VREx | 0.2 | 0 | 2 | 1.518 | 0.049 | 66.99% | 94.61% |
| MTL | 0.0 | 0 | 2 | 1.269 | 0.001 | 79.95% | 96.11% |
| MTL | 0.2 | 0 | 2 | 2.477 | 0.060 | 67.73% | 93.41% |
| SD | 0.0 | 0 | 2 | 0.589 | 0.113 | 85.33% | 98.20% |
| SD | 0.2 | 0 | 2 | 0.930 | 0.262 | 80.44% | 97.60% |
| SD | 0.3 | 0 | 2 | 1.191 | 0.454 | 73.35% | 96.11% |

Table 4: Evaluation of transferability of popular algorithms for domain generalization on Office-Home. **algorithm**: the model that we evaluate; $\delta$: the adversarial radius $\delta$ we choose in Algorithm 1; **max/min index**: the index of the domain with the maximal/minimal (test) classification errors (w.r.t. 0-1 loss); **max/min loss**: the largest/smallest loss among domains (including the target domain); **worst/best acc**: the smallest/largest classification test accuracies among domains (including the target domain).

| algorithm | $\delta$ | max index | min index | max loss | min loss | worst acc | best acc |
|-----------|----------|-----------|-----------|----------|----------|-----------|----------|
| ERM | 0.0 | 0 | 2 | 2.688 | 0.054 | 54.43% | 88.16% |
| ERM | 0.1 | 0 | 2 | 3.701 | 0.098 | 47.63% | 87.37% |
| GroupDRO | 0.0 | 0 | 2 | 2.940 | 0.072 | 58.76% | 88.61% |
| GroupDRO | 0.1 | 0 | 2 | 4.042 | 0.147 | 50.72% | 86.81% |
| SagNet | 0.0 | 0 | 2 | 2.030 | 0.055 | 56.08% | 87.94% |
| SagNet | 0.1 | 0 | 2 | 2.316 | 0.071 | 54.02% | 88.05% |
| Mixup | 0.0 | 0 | 2 | 1.657 | 0.051 | 60.62% | 90.76% |
| Mixup | 0.1 | 0 | 2 | 2.074 | 0.075 | 53.40% | 90.08% |
| CORAL | 0.0 | 0 | 2 | 1.878 | 0.043 | 59.79% | 89.06% |
| CORAL | 0.1 | 0 | 2 | 2.111 | 0.053 | 56.70% | 88.73% |
| MMD | 0.0 | 0 | 2 | 2.201 | 0.037 | 56.49% | 89.74% |
| MMD | 0.1 | 0 | 2 | 2.860 | 0.060 | 50.93% | 88.16% |
| VREx | 0.0 | 0 | 2 | 1.926 | 0.207 | 55.46% | 85.46% |
| VREx | 0.1 | 0 | 2 | 2.414 | 0.245 | 49.28% | 84.89% |
| MTL | 0.0 | 0 | 2 | 2.736 | 0.047 | 52.58% | 87.71% |
| MTL | 0.1 | 0 | 2 | 3.921 | 0.109 | 42.06% | 85.12% |
| SD | 0.0 | 0 | 2 | 1.535 | 0.047 | 64.33% | 91.54% |
| SD | 0.1 | 0 | 2 | 1.717 | 0.049 | 63.51% | 92.33% |