# OpenReview forum: "Quantifying and Improving Transferability in Domain Generalization"
_NeurIPS.cc/2021/Conference — NeurIPS 2021 Poster_

### Official Review · Reviewer_AD3E · 2021-07-13

**Rating:** 7
**Confidence:** 4

**Summary:**

Authors introduced the notion of transferability between pairs of domains (i.e. joint distributions over input x output spaces) along with a discrepancy measure, which can be used to assess how dissimilar are two data sources relative to a given class of predictors. Such a quantity may be useful in either assessing how well is a trained model likely to perform on new data, or as a training signal so that more preference is given to predictors able to generalize out of the training data distribution. The main strength of this work in my opinion is the fact that the proposed transferability measure accounts for the joint distributions over data and labels. However, while I appreciate the depth of the discussion and results in terms of the properties of the proposal, I believe a key missing discussion is the effect that commonly used sets of assumptions have in the proposed transferability measure. Moreover, the evaluation lacks in evidencing the benefits of the proposal relative to current strategies. Please refer to the following for further details.

**Limitations And Societal Impact:**

Yes.

**Main Review:**

Strengths:

+The proposed discrepancy measure accounts for the joint distribution over data and labels, which seems to remove the need for assumptions such as covariate shift, required in common alternatives that only account for the data marginal distribution.

+Authors present a comprehensive set of results and in-depth discussion of the properties of the proposed measure.

+An interesting evaluation is carried out where transferability is estimated for features obtained under different settings appearing in recent literature.

+Algorithm 2 requires a single extra classifier to estimate the worst-case pairwise discrepancy (h’). Past similar approaches use N or even N^2 extra classifiers, where N is the number of training domains.

Weaknesses/suggestions:

-The write-up lacks in discussing the effects of different kinds of distribution shifts in the transferability measures. I would expect to see a discussion similar to [1] where the effect of, for instance, conditional shifts would be analyzed. It seems severe conditional shift, c.f. example in figure 1 in [1], would make the transferability measure explode, which would result in no advantage over more standard discrepancy measures over data marginals only such as the H-divergence. Please clarify whether that’s indeed the case.

-Regarding specifically the empirical evaluation, it lacks in evidencing the effectiveness of the proposed measure since it focuses on analyzing the transferability of features obtained under different algorithms. It would be important to see evidence showing that the estimation errors are not significantly affecting results in practice. As a suggestion, it seems all evaluations are carried out in datasets where covariate shift can be assumed (i.e. only data marginals shift). In that case, we would expect the proposed transferability to correlate with, for example, the H-divergence. Verifying that would serve as evidence that the measure and the algorithm used to implement it actually yield informative results, and are not relevantly affected by, for example, sub-optimality of gradient ascent.

-Another missing piece of evidence in my opinion is in how indicative of generalization the proposed transferability measure is. From a practical perspective, it would be relevant to see evidence that lower transferability implies better generalization.

-Finally, the claim that the proposed transferrability measure is helpful in allowing the analysis to shift from the marginals to the joint distributions should be supported by empirical evidence as well. Are there cases where more standard notions such as the H-divergence wouldn't work and the proposal would?

Minor suggestions:

Since transferability is a property of a pair of domains and a hypothesis class, perhaps that should be indicated in the notation; i.e. (\delta_S, \delta_T, H)-transferable should be used instead of (\delta_S, \delta_T)-transferable.

[1] Zhao H, Des Combes RT, Zhang K, Gordon G. On learning invariant representations for domain adaptation. International Conference on Machine Learning 2019 May 24 (pp. 7523-7532). PMLR.

Post rebuttal edit:

I went through the authors' responses which clarified the effect that conditional shift would have on the proposed measure. I decided to update my score to 'accept'. I do like the idea since it's the first domain-divergence measure that I'm aware of that accounts for the joint distribution over data and labels, as opposed to only one of the marginal distributions.

One concern that remains to me is the coverage of the evaluation, which I think is very limited. While the main strength of the approach, in theory, is the fact that it accounts for the joint distributions, all evaluations are carried out on cases where only the data marginals shift. Moreover, only the proposed transferability is evaluated, and it's not compared with more standard divergence measures such as the H-divergence. If the proposal yields meaningful domain divergence results, we would expect a strong correlation between their proposal and the standard measures in the considered evaluations, and that would give confidence that the proposal works in practical settings. For example, on PACS, pairwise H-divergences could be compared with the proposed transferability, and ideally, they would correlate more or less strongly depending on the label marginals shift.

In summary, I think the idea is very interesting, novel, and solves an important issue, the theoretical results are strong and insightful, but the evaluation is limited. All in all, I think the merits of the paper outweigh its flaws.

**Time Spent Reviewing:**

6

---

> ### Author Response · Authors · 2021-08-10
> **Response to Reviewer 4**
>
> We would like to thank the reviewer for a careful reading of our paper and a nice summary of our strengths and weaknesses. Also, thank you for your suggestions on how to improve our current work. Please find our point-by-point responses below:
>
> > **Q1: I would like to see a discussion on the effects of different kinds of distribution shifts in the transferability measures. It seems severe conditional shift c.f. example in Figure 1 in Zhao et al would result in no advantage over more standard discrepancy measures over marginals such as H-divergence. Please clarify.**
>
> A1: Thank you for bringing up this comparison. We did discuss label shift and cite Zhao et al ICML 2019 (your ref [1]) in our paper, see Line 46 and ref [51]. Note that the goal of Figure 1 in [51] is to show that even if the $H$-divergence is 0, any classifier well-trained on the source domain cannot generalize to the target. From this perspective, $H$-divergence is not a good measure of direct transferability. As a comparison, in this example, our transferability measure is also 1, hence it is consistent with the fact that under severe conditional shift, any source classifier cannot transfer to a target domain directly.
>
> Moreover, Zhao et al. [51] shows that the optimal joint error can be large under label shift. We also checked the label shift in our experiments in the following table following [51]:
>
> | total variation | A | C  | P | S
> --- | --- | --- | --- | ---
> | **A** | 0.0 | 0.12 | 0.11 | 0.3
> | **C** | 0.12 | 0.0 | 0.18 | 0.24
> | **P** | 0.11 | 0.18 | 0.0 | 0.37
> | **S** | 0.3 | 0.24 | 0.37 | 0.0
>
> In this table, we compute the total variation distance between different pairs of domains (A for art painting, C for cartoon, P for photo, S for sketch) in the PACS dataset. Note that the range of total variation is between $0$ and $1$. From the conclusions in the aforementioned [51], we may not apply $H$-divergence in this case due to this label shift.
>
> > **Q2: It would be important to see evidence showing that the estimation errors are not significantly affecting results in practice.**
>
> A2: See our response to Q4 below.
>
> > **Q3: It seems all evaluations are carried out in datasets where covariate shift can be assumed and we would expect the proposed transferability to correlate with, for example, the $H$-divergence.**
>
> A3: Thank you for your suggestion. Our Figure 2 is an example of covariate shift because the conditional distributions are the same. Due to the severe shift of marginal distributions, $H$-divergence is large. However, our transfer measure is still small. Therefore, even under this covariate shift assumption, our transfer measure is advantageous over $H$-divergence. See also our Prop. 3 and the target error bound with $H$-divergence [7].
>
> > **Q4: ...are not relevantly affected by, for example, sub-optimality of gradient descent.**
>
> A4: In practice we only need to show that there is a gap between source and target accuracies for many algorithms by doing maximization. We can get a larger gap if we optimize better, but our current optimization already shows there is a large gap, even though it might be sub-optimal. In fact, the sub-optimality of gradient descent is a perennial theoretical question for deep learning and there is no easy answer for it.
>
> > **Q5: It would be relevant to see evidence that lower transferability implies better generalization.**
>
> A5: At least in theory, our Prop 3 indeed shows that lower transferability implies better generalization. In practice, our Alg 2 shows that by minimizing the transfer measure we can obtain better generalization to the target domain, as supported by our experiments shown in Fig 3.
>
> > **Q6: Are there cases where more standard notions such as the $H$-divergence wouldn't work and the proposal would?**
>
> A6: We have cases from both theory and experiments. See our answers to Q1/A1 and Q3/A3. In particular, our Figure 2 and Figure 1 in [51] serve as simple examples.
>
> > **Q7: Perhaps the hypothesis class should be indicated in the notation.**
>
> A7: Thank you for the suggestion. We will modify the draft accordingly.

---

> ### Author Response · Authors · 2021-08-25
> **Thank you for your update**
>
> Thank you for going through our rebuttal and updating your score in time. We appreciate your careful follow-up and constructive suggestions that we will surely incorporate to the revised version. We will improve the evaluation by comparing our transfer measures with H-divergence in more detail, and illustrate the two marginal shifts in theory and experiments. (see Q1/A1 and Q3/A3 in our previous rebuttal)

---

### Official Review · Reviewer_RBZG · 2021-07-15

**Rating:** 7
**Confidence:** 5

**Summary:**

This paper aims to provide a formulation of transferability. The main intuition is that any near-optimal source classifier should be also near-optimal on the target domain. So the authors quantify transferability by measuring the relation between near-optimal classifiers in source and target domains and derive genralization bounds based on this. In addition, new algorithm is proposed based on the intuition.

**Limitations And Societal Impact:**

Not applicable.

**Main Review:**

Originality: The problem of quantifying transferability is not well studied and the reviewer agrees that proposing a formulation of transferability or invariance is important and novel (though there are many other concurrent papers doing this, e.g., https://arxiv.org/abs/2008.01883, https://arxiv.org/abs/2106.04496, which might be better to be included later)

Quality: The basic formulation of transferability in this paper is like a relaxation of the IRM objective with quantifiable surrogates. The relaxation is reasonable though the quantitative part is similar to the DANN-style discrepancy. Given the improvements on the empirical performance compared to DANN and other distributional matching baselines, the reviewer tends to vote for accept.

Clarity: The writing of this paper is clear.

Significance: The formulation and algorithm proposed in this paper are similar to the previous work in the form but bring new thoughts and improvements to the area.

**Time Spent Reviewing:**

2

---

> ### Author Response · Authors · 2021-08-10
> **Response to Reviewer 3**
>
> Thank you for your encouraging comments and your positive feedback! We would like to clarify some points more carefully:
>
> > **Q1: The basic formulation of transferability in this paper is like a relaxation of the IRM objective.**
>
> A1: We agree that our work can be related to IRM. However, the principle of IRM is invariant optimal predictors, while the principle of our method is the invariance of excess risk. This difference might seem slight, but it is crucial. Our analysis shows that the invariance of excess risk is exactly what the algorithm needs in order to guarantee generalization to the target domain. As a comparison, IRM asks the optimal predictors to be the same across domains, but it does not mention any error by itself. Empirically, our method also obtains better performance than IRM. See RotatedMNIST for a comparison in Fig 3.
>
> > **Q2: The proposed method shares some similarity with DANN-style discrepancy.**
>
> A2: We agree that our theory has a connection with DANN-style discrepancy, which is discussed in lines 119-123 and Appendix B.5. The main difference is that our formulation uses *joint distributions* rather than *marginals*, which marks an important step that goes beyond existing DANN-style discrepancies. Moreover, Example 8 shows that even if two distributions (source and target) do not match, it could still be transferable, and our transfer measure describes the degree of transferability in this example. Note that we cannot use DANN-style discrepancy in this case because the marginal distributions on $X$ differ a lot ($|p_S(X) - p_T(X)| = 0.8$ for $-1 \leq X < 1$ if we do the computation).
>
>
> > **Q3: Some more references can be added.**
>
> A3: We thank the reviewer for suggesting some recent interesting and related works, and we will add them to a revised version. The paper arxiv:2008.01883 by Koyama and Yamaguchi is related and it shares a similar idea as IRM [3] using invariant predictors. The paper arxiv:2106.04496 is a concurrent work that we were not aware of it when preparing for this work. One of the differences is that this paper focuses on proposing a model selection rule given their designed criterion and instead we are concerned with a formal measure of transferability.

---

### Official Review · Reviewer_NJSd · 2021-07-17

**Rating:** 6
**Confidence:** 3

**Summary:**

This paper defines a notion of "transferability" between features from different domains, which is different from common distribution discrepancy measures such as total variation and Wasserstein distance. This measure of transferability is measured with labeled samples from both domains, and provides a bound on the target error. The empirically test transferability of the features learned by current domain generalization algorithms, and some don't do that well in this transferability measure. They propose an algorithm based on optimizing the measure of transferability across training domains and test it on some domain generalization datasets, including a satellite dataset.

**Limitations And Societal Impact:**


The authors mention that their algorithm requires a large number of inner maximization steps, increasing training time, and that moderate hyperparameter tuning is needed for additional hyperparameters. In terms of broader impact, the authors warn that just because an algorithm passes their transferability check, it does not guarantee good domain generalization performance.

**Main Review:**

Quality

The general structure of the work seems to be to find a metric (transferability) that plugs into a generalization bound, then convert this into an algorithm by optimizing the metric. This paper gives some interesting results about source vs target accuracy when subject to parameter space pertubations, but these comparisons are somewhat better suited for their method since they optimize something related to this directly. They give a comprehensive theoretical analysis including finite sample results. The main concerns are in the formulation and the empirical results. For formulation, it's not immediately clear what this method leverages better than other domain generalization methods since it also only uses the given training domain information for enforcing invariance - is the bound tighter than others, etc.? More intuition on this could help. For empirical results, from Figure 3 it seems that the SD baseline is comparable to the proposed algorithm in target accuracy across all the datasets. Also, error bars aren't given in the plots and the results could be within noise.

- As the authors mention in line 258, the measure of transferability depends on labeled samples from the target domain. They circumvent this in the algorithm by optimizing transferability between a number of source domains. However, there's a bit of a conceptual gap here, since it is straightforward to say that we optimize a bound on the target accuracy of a particular target distribution that we measure transferability on, but it's less clear how optimizing for transferability on a number of source domains will generalize to a new target.
- In line 242 and equation 13, $\Gamma$, the $\delta_S$-minimal set of $\epsilon_S$, is replaced by a $\delta$-ball in parameter space. This relies on an assumption on the neural network to be Lipschitz in the parameter space. However, for wide neural networks, it seems that small perturbations in parameter space can cause very large changes. This could make the bound in equation 13 very loose. It's also unclear that comparisons wrt to the $\delta$-ball in parameter space like in Figure 3 are valid due to the unknown (and possibly very small) range of valid $\delta$. The overall finding that target accuracy decreases much faster with parameter space perturbations is interesting, however.
- In section 2.1, the paper states that the defined metric is different from IPMs since it relies on an underlying function class, but this is exactly what the "neural network distance" IPMs used in theory of GANs do as well [1].

- Minor: in the self-driving example in the intro, night/day shouldn't be a shift where we should be applying domain generalization techniques, since it's so easy to collect data in both the night and day. Time shift might be more appropriate here (you can't collect data from the future).

Significance

Domain generalization methods generally don't seem to work much better than ERM, according to some empirical works, so advancements in this field are necessary. The overall approach in this paper is nice and comes from a principled point of view, however the empirical results are inconclusive whether this paper's method improves upon the other baselines.

Originality

The final algorithm seems related, though different, to invariance based methods such as IRM or group DRO. The principle seems to be to make the errors on every training domain to be the same within a ball in the parameter space (learn parameter-stable networks). The ball in parameter space seems to be a limitation of the technique / inability to characterize the $\delta$-minimal set of hypothesis, however, and not necessarily fundamental. The measure of transferability is nice, although it requires labeled data in both source and target so it isn't necessarily surprising (whereas using only unlabeled target data to estimate transferability would be tougher).


Clarity

The paper could be structured more hierarchically - it dives into math as part of the essential explanation. I think the clarity could benefit from giving a high level overview and then going into the details.
- Fig 1: from the description of transferability, it's unclear why we're considering $\delta$ perturbations in the parameter space. This is only introduced in Section 4.
- Def 2 is named quantifiable transfer measures, but I believe only the third one is computable since it's the only one that doesn't require knowing optimal errors. Whether something should be considered quantifiable if it is not computable is unclear.
- Since there are 3 transfer measures defined, it's a bit hard to keep track of which one is being used at the moment, and what the purpose of each of the measures are. Perhaps the paper could state which one will be the main one to work with, and then using these propositions we can make it computable.
- In Fig 3, the brown color is used for many different algorithms, consider deduplicating the color.


[1] Sanjeev Arora, Rong Ge, Yingyu Liang, Tengyu Ma, Yi Zhang. Generalization and Equilibrium in Generative Adversarial Nets (GANs), 2017.

**Time Spent Reviewing:**

3

---

> ### Author Response · Authors · 2021-08-10
> **Response to Reviewer 2**
>
> We would like to thank the reviewer for the detailed and constructive comments. We sincerely appreciate all of them, and we will surely incorporate them to further improve the presentation of our work. In what follows we provide detailed responses to the questions raised by the reviewer:
>
> > **Q1: These comparisons are somewhat better suited for their method since they optimize something related to this directly.**
>
> A1: Note that in our Algorithm 2, we are only optimizing the transfer measure between source domains, but our evaluation method in Algorithm 1 includes the target domain as well.
>
> > **Q2: Is the bound tighter than others, etc.?**
>
> A2: Thank you for the great question. Our bound is indeed tighter than other bounds such as $H$-divergence [7]. It is also more general since $H$-divergence only works for binary classification and only focuses on the marginal feature distributions. To prove that our bound is tighter, we can take $h^*\in \textrm{argmax}_{h\in \Gamma} \epsilon_T(h) - \epsilon_S(h)$, and plug it into the theorem for $H$-divergence in [7].
>
> > **Q3: From figure 3 it seems that the SD baseline is comparable to the proposed algorithm across all the datasets in the target accuracy across all the datasets.**
>
> A3: For the OfficeHome dataset, our method is in fact better than SD as shown in the following table, which shows he improvement of our method over SD (in percentage) for the OfficeHome dataset:
>
> | target acc (%) | $\delta = 0$ | $\delta = 0.05$ | $\delta = 0.1$ | $\delta = 0.15$
> --- | --- | --- | --- | ---
> | SD                  | 64.33           | 63.71                | 63.51              | 50.1
> | Ours | 65.15 | 65.15 | 63.71 | 55.67
> **Improvement** | **+0.82** | **+1.44** | **+0.20** | **+5.57**
>
>  Moreover, in the PACS dataset our Transfer algorithm also improves over SD by +7.3\% in terms of the target accuracy at $\delta = 0.3$. For the WILDSFMoW dataset, our method improves over SD by $+1.2\%$ for the unperturbed target accuracy at $\delta = 0$, which is a standard metric in domain generalization. However, in some cases SD is indeed better.
>
>
> > **Q4: Error bars aren't given in the plots and the results could be within noise.**
>
> A4: We would like to bring to the reviewer's attention that we performed several different trials (see lines 868-870 in the appendix) of different methods. But instead of computing the variance and showing error bars, we choose to study the worst case directly, which reflects *better* the accuracy gap in Algorithm 1 between the source and the target. We can also compute the standard deviation as the reviewer suggests. As an illustrating example, we compute the target accuracy for our Transfer algorithm in the following table, which shows the target accuracies at different parameter deviations for our Transfer algorithm on RotatedMNIST:
>
> | $\delta$ | 1.0 | 2.0 | 3.0 |
> --- | --- | --- | ---
> | target acc (%) | $95.89 \pm 0.03$ | $94.10 \pm 0.08$ | $89.41 \pm 0.30$
>
> > **Q5: It is less clear how optimizing the transferability on a number of source domains will generalize to a new target.**
>
> A5: Thanks for your question. Please check Theorem 12, which partly addresses this question. In domain generalization, since we have no information about the target domain (not even unlabeled data), we need to assume that the target domain shares some properties with a series of source domains. The convex hull assumption in Theorem 12 naturally reflects this. With this assumption, we can theoretically upper bound the target error in Eq. (16). The implementation of our Alg 2 in the experimental section also gives evidence that this assumption is practical: by optimizing the problem in Eq. (15) we can obtain a high target accuracy.
>
> > **Q6: For wide neural networks it seems that small perturbations in parameter space can cause very large changes, and the Lipschitz constant can be large. This could make the bound in equation 13 very loose. It's also unclear that comparisons wrt to the $\delta$-ball in parameter space in Figure 3 are valid due to the unknown range of valid $\delta$.**
>
> A6: In our experiments we choose all the classifiers to be linear (line 863), therefore the aforementioned Lipschitz constant only depends only on the loss function and the classifier, but not wide neural networks that appear in the feature embedding, see eq. (10).
>
> In practice we find the valid $\delta$ through cross-validation. This can be efficient if we perform binary search of $\delta$ and see the decrease of the target/source accuracies. As long as the range of $\delta$ can reflect the gap of accuracies between target and source as in Figure 3, we do not see a problem here.
>
> > **Q7: In Sec 2.1 the paper states that the defined metric is different from IPMs, but this is exactly Arora et al 2017 did in theory of GANs.**
>
> A7: Our realizable transfer measure is indeed related to Arora et al. and we are happy to cite the reference. But it is different from other commonly-used IPMs such as MMD, Wasserstein distance, Dudley metric and Kolmogorov distance (line 142-144) since there is a new function class involved. Please check also Appendix B.2 for a more detailed discussion in this regard.
>
> > **Q8: (Minor) Night/day shouldn't be a shift since it's so easy to collect data in both the night and day.**
>
> A8: Sure. We will change the example to driving in different cities/regions.
>
> > **Q9: The empirical results are inconclusive if this paper's method improves upon the other baselines.**
>
> A9: Please see Q3 and A3.
>
> > **Q10: The ball in parameter space seems to be a limitation to characterize the $\delta$-minimal set.**
>
> A10: Please see Q6 and A6.
>
> > **Q11: Def 2 is named quantifiable transfer measures but I believe only the third one is computable.**
>
> A11: Theoretically, yes. However, although it is hard to exactly compute the optimal errors in deep learning, we can use the smallest errors we found as surrogates in practice. Therefore we can approximate these transfer measures, see also line 243.
>
>
> > **Q12: It is a bit hard to keep track of which transfer measure is being used at the moment and the purpose of each measures. Perhaps the paper could state which one will be the main one to work with.**
>
> A12: Thanks for this suggestion. We will follow your suggestion in the next iteration of our paper. These three transfer measures are all necessary ingredients of our paper. They are closely related (see eq. (3) and Prop 4) and we use different measures for different purposes. The first one in Eq. (2) is the weakest measure and it directly relates to Def 1 via Prop 5. If we want two domains to be transferable to each other, we symmetrize Eq. (2) to obtain the second transfer measure in eq. (3), which is also equivalent to Def 1. This symmetric transfer measure reduces to Eq. (4) if the optimal errors are zero. To show the relation with IPMs, we need the realizable version in Sec 2.1.
>
> > **Q13: In Fig 3 the brown color is used for many algorithms.**
>
> A13: Thank you for the suggestion. We will modify the figure.

---

> > ### Comment · Reviewer_NJSd · 2021-09-02
> > **response**
> >
> > Thanks for the response, which addresses some concerns about the experimental results and theoretical setup. Overall, the experiments are a bit limited, but my previous concerns about the existing experiments are less. The paper should also be written more clearly/hierarchically, as mentioned in the review.
> > However, I'm not sure how Q6/A6 answers Q10?
> > After the author response, I increase my score to 6.

---

> > > ### Author Response · Authors · 2021-09-02
> > > **Thank you for your update**
> > >
> > > Thank you for your time for checking our rebuttal and updating your score, and again for your detailed and constructive comments. We are glad that some of your concerns have been addressed. We will surely improve our presentation and incorporate all your comments in the revised version. We would like to explain more on Q10. It is true that our parameter ball is a sufficient condition for the $\delta$-minimal set but may not be necessary if $h$ is a neural network. However, we can use the evaluation on the parameter ball to lower bound our transfer measures, as we explained in line 240 and line 246-250. To fully characterize the $\delta$-minimal set, we resort to functional norm balls, see eq. (9) and Appendix B.4. These functional norm balls can be further characterized with parameter balls if the classifier $h$ is a linear function.

---

### Official Review · Reviewer_hkRr · 2021-07-18

**Rating:** 5
**Confidence:** 3

**Summary:**

This paper aims at identifying and quantifying transferability in learning algorithms. The paper proposes a new upper bound on the target error. The proposed algorithm is an adversarial optimisation problem that empirically proved to be effective.

**Limitations And Societal Impact:**

All in all, I believe the technical quality of the paper may be good but poorly presented.

**Main Review:**

The paper tackles an interesting and important problem. However, the main issue I have with the paper is that it is hard to read and barely provides intuitive explanations on the results. It is also a very notation-heavy paper, but they seem to be used without much introduction and discussion. The paper could significantly benefit from a section on problem setup and notations. For instance, before section 4, z is used but it is introduced in Section 4. Also, S_i is not introduced, I am assuming it is one of the source domains.

In addition, the main contribution seems to be to draw connection between IPM measures and the transferability that is already known in the literature. The results section is also poorly presented and is not clear what the experimental setup is. It is also not better explained in the supplements.

I have some concerns about the problem set up and the overclaim of the paper. I understand though that this paper follows similar ones in the domain transfer literature, but I believe for more modern deep learning settings it is not sufficient. It would be interesting to see how these bounds be different if the i.i.d assumption is removed and a more realistic setting is used. In fact, I believe the main obstacle in using such bounds is that the main question in transferability is how to address these confounders. For instance, if we had a dataset of cats vs dogs where all the cats have a blue background and all the dogs a red one, we expect with the simplicity bias in neural networks, the model focuses on the background alone. Therefore the transferability to another problem with colour as the main discriminative signal is intuitively easier than to a cat vs dog dataset where images are taken in the wilds.

With the rise of self-supervised and pre-training, I believe this paper could be more impactful if it included discussions on those lines of work. They aim at learning the invariances and that is how they achieve better generalisation.


**Time Spent Reviewing:**

4

---

> ### Author Response · Authors · 2021-08-09
> **Response to Reviewer 1**
>
> Thank you for your time devoted to providing the valuable feedback. It seems that the main concern is the presentation of our technical results. We will surely address this by providing extended discussions and intuitive explanations to the results presented in the paper in the next version of the paper. Please see below for our responses to the detailed questions.
>
> > **Q1: The paper is hard to read and barely provides intuitive explanations.**
>
> A1: Thanks for the comments. As we mentioned before, we will strive to improve the presentation of our work. However, we do try to provide intuitive explanations throughout the paper, e.g., line 66-69, line 119-123, line 156-162, line 191-194, line 268-270, and line 276-283. These comments mainly follow the introduction of our technical results. That being said, we notice that some explanations are still missing, and we will improve the presentation through the following:
> * For Def 1 we will elaborate more on the $\delta$-minimal set which represents the near-optimal set of classifiers as explained in line 95. For Def 2 we will elaborate more on the meaning of the excess risk, which is the relative error compared to the optimal error.
> * We will provide more discussions for Prop 11 as well. For example, one direct implication is that if the transfer measure is small, then a near-optimal classifier of the surrogate loss in the source domain would be near-optimal in the target domain for the 0-1 loss.
>
> >  **Q2: The notations are heavy and you could add a section on problem setup and notations.**
>
> A2: Thanks for the great suggestion! We will definitely add one paragraph to properly introduce the notation. Our setting is introduced between lines 76-88 in the manuscript and we will add a title to make it clearer.
>
> > **Q3: Before Section 4, $z$ is used but it is introduced in Section 4.**
>
> A3: We searched the draft and did not find the $z$ that you meant. The only place we found is in Line 198, Theorem 10, where $z$ has the same meaning as $x$ in Line 142. We will fix this typo.
>
> > **Q4: $\mathcal{S}_i$ is not introduced**
>
> A4: We introduced $\mathcal{S}_i$ in Line 77.
>
> > **Q5: The main contribution seems to be to draw connection between IPM and the transferability that is already known in the literature.**
>
> A5: To the best of our knowledge, we are not aware of any literature that defines transferability as we do in Definitions 1 and 2. We would appreciate it if the reviewer could kindly point out the paper that formally introduced this notion. In fact, this formal definition of transferability is one of our main contributions, together with an algorithmic framework to approximate the transfer measures.
>
> > **Q6: It is not clear what the experimental setup is. It is also not better explained in the supplements.**
>
> A6: Please find the detailed experimental setup in Appendix C.2, including hardware, batch size, optimization (learning rates, optimizer), data split, neural architecture, training steps, and so on.
>
> > **Q7: For more modern deep learning settings it is not sufficient. It would be interesting to see how these bounds be different if i.i.d. assumption is removed.**
>
> A7: To our knowledge, most transfer learning theory focuses on i.i.d. samples respectively from target and source domains such as refs [7, 51], despite the distribution shift among domains. We agree that generalizing to non-i.i.d. settings is interesting but it remains a great challenge and is out of the scope of our paper.
>
> > **Q8: I believe the main question in transferability is how to address these confounders. With colour as the main discriminative signal the problem the transferability is intuitively easier.**
>
> A8: Thank you for bringing up this interesting perspective. Indeed we agree that tackling out-of-distribution generalization from a causal perspective is an interesting and promising direction, but it often requires prior knowledge about the underlying generative process of the data, which is usually not available in practice. Instead, in this paper we focus on defining transferability analytically and developing practical methods to approximate it. On the other hand, there are recent related works such as IRM [3] that tackles the problem from a causal perspective, and GroupDRO [36] that uses spurious association, We compare our Algorithm 2 with these methods in Figure 3.
>
>
> > **Q9: This paper could be more impactful if it includes discussions on self-supervised learning and pre-training.**
>
> A9: Thank you for your suggestions. The recent line of work on self-supervised learning and pre-training is definitely exciting and interesting. However, we'd like to draw the reviewer's attention that our focus here is on evaluating and improving transferability of feature learning methods in the field of *domain generalization*. Given the large and diverse field of representation learning, it is almost impossible to exhaust all the marginally related subfields. However, we are happy to discuss some recent related works on SSL such as SimCLR, MoCo, BYOL, etc. in the next iteration of our paper:
> * SimCLR: Chen et al ICML 2020, A simple framework for contrastive learning of visual representations.
> * MoCo: He et al, Momentum contrast for unsupervised visual representation learning, CVPR 2020.
> * BYOL: Grill et al, Bootstrap your own latent - a new approach to self-supervised learning, NeurIPS 2020.

---

### Decision · Program_Chairs · 2021-09-27

**Decision:**

Accept (Poster)

**Comment:**

This paper focuses on transferability in context of OOD generalization. The authors define the term, point out connection and differences with current discrepancy methods and then prove that this measure can be transferability can be estimated with enough samples and give a new upper bound for the target error based on our transferability. Empirically, they evaluate the transferability of the feature embeddings learned by existing algorithms for domain generalization. They also propose a new method for learning transferable features and test it over some benchmarks. The idea is interesting and the theoretical investigations are worthwhile. The authors addressed the questions and concerns raised by reviewers and I believe their responses are satisfactory. I request the authors to update the paper as promised in the discussion period and add the requested information and clarifications as well as comparison with the related works raised by the reviewers.